

# Recalibrating Decadal Climate Predictions

# –

# What is an adequate model for the drift?

Alexander Pasternack[1], Jens Grieger[1], Henning W. Rust[1], and Uwe Ulbrich[1]

[1]Institute of Meteorology, Freie Universität Berlin, Berlin, Germany

**Correspondence:** A. Pasternack (alexander.pasternack@met.fu-berlin.de)

**Abstract.** Near-term climate predictions such as decadal climate forecasts are increasingly being used to guide adaptation measures. To ensure the applicability of these probabilistic predictions, inherent systematic errors of the prediction system must be corrected. In this context, decadal climate predictions have further characteristic features, such as the long time horizon, the lead-time dependent systematic errors (drift) and the errors in the representation of long-term changes and variability. These

features are compounded by small ensemble sizes to describe forecast uncertainty and a relatively short period for which typically pairs of re-forecasts and observations are available to estimate calibration parameters. With *DeFoReSt* (Decadal Climate Forecast Recalibration Strategy), Pasternack et al. (2018) proposed a parametric post-processing approach to tackle these problems. The original approach of *DeFoReSt* assumes third order polynomials in lead time to capture conditional and unconditional biases, second order for dispersion, first order for start time dependency. In this study, we propose not to restrict

orders a priori but use a systematic model selection strategy to obtain model orders from the data based on non-homogeneous boosting. The introduced *boosted recalibration* estimates the coefficients of the statistical model, while the most relevant predictors are selected automatically by keeping the coefficients of the less important predictors to zero. Through toy model simulations with differently constructed systematic errors, we show the advantages of *boosted recalibration* over *DeFoReSt*. Finally, we apply *boosted recalibration* and *DeFoReSt* to decadal surface temperature forecasts from the MiKlip Prototype

system. We show that *boosted recalibration* performs equally well as *DeFoReSt* and yet offers a greater flexibility.

## 1   Introduction

Decadal climate predictions focus on describing the climate variability for the coming years. Significant advances could be achieved by recent progress in model development, data assimilation and climate observation. A need for up-to-date and reliable short-term climate information for adaptation and planning accompanies this progress (e.g., Meredith et al., 2018). In this

context, international (e.g., DCPP and WCRP grand challenge) and national projects like the German initiative Mittelfristige Klimaprognosen (MiKlip) have developed model systems to produce a skillful decadal ensemble climate prediction (Pohlmann et al., 2013a; Marotzke et al., 2016). Typically, climate predictions are framed probabilistically to address the inherent uncertainties caused by imperfectly known initial conditions and model errors (Palmer et al., 2006).



Despite the progress being made in decadal climate forecasting, such forecasts still suffer from considerable systematic errors like unconditional, and conditional biases and ensemble over- or underdispersion. Those errors generally depend on forecast lead-time since models tend to drift from the initial state towards its own climatology. Furthermore, there can be a dependency on initialization time when long term trends of the forecast system and observations differ (Kharin et al., 2012). In this regard, Pasternack et al. (2018) proposed a Decadal Forecast Recalibration Strategy (*DeFoReSt*) which accounts for the three above mentioned systematic errors. While DCPP recommends to calculate and adjust model bias for each lead time separately to take the drift into account, Pasternack et al. (2018) uses a parametric approach to describe systematic errors as a function of lead time. *DeFoReSt* uses third order polynomials in lead time to capture conditional and unconditional biases, second order for dispersion and a first order polynomial to model initialisation time dependency. Third order polynomials for the drift have been suggested by Gangstø et al. (2013) and have later been used by Kruschke et al. (2015). Hence, *DeFoReSt* is an extension of the drift correction approach proposed by Kruschke et al. (2015), accounting also for conditional bias and adjusting the ensemble spread. The associated *DeFoReSt* parameters are estimated by minimization of the CRPS, analog to the nonhomogeneous Gaussian regression approach by Gneiting et al. (2005).

Although *DeFoReSt* with third/second order polynomials turned out in past applications to be beneficial for both, full field initialized decadal predictions (Pasternack et al., 2018) and anomaly initialized counterparts (Paxian et al., 2018), as well as decadal regional predictions (Feldmann et al., 2019), it is worthwhile challenging the a priori assumption by using a systematic model selection strategy. In this context, full field initializations show larger drifts in comparison to anomaly initializations even though drift of the latter is not negligible, particularly when taking initialization time dependency into account (Kruschke et al., 2015).

For post-processing of probabilistic forecasts with non-homogeneous Gaussian regression Messner et al. (2017) proposed the non-homogeneous boosting to automatically select the most relevant predictors. Originally, boosting has been developed for automatic statistical classification (Freund and Schapire, 1997), but has been used as well for statistical regression (e.g. Friedman et al., 2000; Bühlmann and Yu, 2003; Bühlmann et al., 2007).

Unlike other parameter estimation strategies based on iterative minimization of an objective function by simultaneously updating the full set of parameters, boosting only updates one parameter at a time; the one that leads to the largest decrease in the objective function. As all parameters are initialized to zero, those parameters corresponding to terms which do not lead to a considerable decrease in the objective function – hence are not relevant – will not be updated and thus will not differ from zero; the associated term has thus no influence in the predictor. Here, we extend the underlying non-homogeneous regression model of *DeFoReSt* to higher order polynomials and use boosting for parameter estimation. Additionally, cross-validation identifies the optimal number of boosting iteration and serves thus for model selection. The resulting boosted non-homogeneous regression model is hereafter named *boosted recalibration*.

A toy model producing synthetic decadal forecasts-observation pairs is used to study the effect of using higher order polynomials and boosting on recalibration. Moreover, we compare *boosted recalibration* and *DeFoReSt* to recalibrate forecasts from the *MiKlip* decadal prediction system.





The paper is organized as follows: Sec. 2 introduces the MiKlip decadal climate prediction system and the corresponding reference data used, Sec. 3 describes the decadal forecast recalibration strategy *DeFoReSt* and introduces *boosted recalibration*,
an extension to higher order polynomials, parameter estimation with non-homogeneous boosting and cross validation for model selection. A toy model developed in Sec. 4 is the basis for assessing recalibration with *boosted recalibration* and *DeFoReSt*. The subsequent Section 5 uses both approaches to recalibrate decadal surface temperature predictions from the MiKlip system. Analogously to Pasternack et al. (2018), we assess the forecast skill of global mean surface temperature and temperature over the North Atlantic subpolar gyre region (60°-10°W, 50°-65°N). The latter has been identified as a key region for decadal
climate predictions (e.g. Pohlmann et al., 2009; van Oldenborgh et al., 2010; Matei et al., 2012; Mueller et al., 2012). Section 6 closes with a discussion.

## 2    Data and methods

### 2.1    Decadal climate forecasts

Basis for this study are retrospective forecasts (hereafter called hindcast) of surface temperature from the Max-Planck-Institute
Earth System Model in a low-resolution configuration (MPI-ESM-LR). The atmospheric component of the coupled model is ECHAM6 at a horizontal resolution of T63 with 47 vertical levels up to 0.01 hPa (Stevens et al., 2013). The ocean component is MPIOM with a nominal resolution of 1.5°. and 40 vertical levels (Jungclaus et al., 2013). This setup together with a full-field initialization of the atmosphere with ERA40 (Uppala et al., 2005) and ERA-Interim (Dee et al., 2011), as well as a full-field initialization of the Ocean with the GECCO2 reanalysis (Köhl, 2015) is called the *MiKlip Prototype System*. This full-field
initialization nudges the full atmospheric or oceanic fields from the corresponding reanalysis to the MPI-ESM, not just the anomalies. A detailed description of the Prototype system is given in Kröger et al. (2018). In the following, we use a hindcast set from the *MiKlip Prototype System* with 50 hindcasts, each with 10 ensemble members integrated for 10 years started every year in the period 1961 to 2010.

### 2.2    Reference data

The Met-Office's Hadley Centre and the Climatic Research Unit at the University of East Anglia produced *HadCRUT4* (Morice et al., 2012), an observational product used here as a reference to verify the decadal hindcasts. The historical surface temperature anomalies with respect to the reference period 1961-1990 are available on a global 5°-by-5° grid on a monthly basis since January 1850. *HadCRUt4* is a composite of the *CRUTEM4* (Jones et al., 2012) land-surface air temperature dataset and the *HadSST3* (Brohan et al., 2006) sea-surface temperature (SST) dataset.

### 85    2.3    Assessing reliability and sharpness

To assess the performance of *boosted recalibration* w.r.t. *DeFoReSt*, we use the same metrics as in Pasternack et al. (2018). For the sake of completeness and readability these are presented in this section again.





Calibration or reliability refers to the statistical consistency between the forecast PDFs and the verifying observations. Hence, it is a joint property of the predictions and the observations. A forecast is reliable if forecast probabilities correspond

to observed frequencies on average. Alternatively, a necessary condition for forecasts to be reliable is given if the time mean intra-ensemble variance equals the mean squared error (MSE) between ensemble mean and observation (Palmer et al., 2006).

A common tool to evaluate the reliability and therefore the effect of a recalibration is the rank histogram or *Talagrand diagram* which were separately proposed by Anderson (1996); Talagrand et al. (1997); Hamill and Colucci (1997). For a detailed understanding, the rank histogram has to be evaluated by visual inspection. Analog to Pasternack et al. (2018), we

use the *Ensemble Spread Score* (ESS) as a summarizing measure. The ESS is the ratio between the time mean intra-ensemble variance $\bar{\sigma^2}$ and the mean squared error between ensemble mean and observation, $MSE(\mu, y)$ (Palmer et al., 2006; Keller and Hense, 2011):

$$ESS = \frac{\bar{\sigma^2}}{MSE(\mu,y)}, \tag{1}$$

with

$$\bar{\sigma^2} = \frac{1}{k}\sum_{j=1}^{k}\sigma_j^2, \tag{2}$$

and

$$MSE(\mu,y) = \frac{1}{k}\sum_{j=1}^{k}(y_j - \mu_j)^2. \tag{3}$$

Here, $\sigma_j^2, \mu_j$ and $y_j$ are the ensemble variance, the ensemble mean and the corresponding observation at time step $j$, with $j = 1, ..., k$, where $k$ is the number time steps.

Following Palmer et al. (2006), $ESS = 1$ indicates perfect reliability. The forecast is overconfident when $ESS < 1$, i.e., the ensemble spread underestimates forecast error. If the ensemble spread is greater than the model error ($ESS > 1$), the forecast is overdispersive and the forecast spread overestimates forecast error. To better understand the components of the $ESS$, we also analyze the mean squared error $MSE$ of the forecast separately.

Sharpness, on the other hand, refers to the concentration or spread of a probabilistic forecast and is a property of the

forecast only. A forecast is sharp, when it is taking a risk, i.e., when it is frequently different from the climatology. The smaller the forecast spread, the sharper the forecast. Sharpness is indicative of forecast performance for calibrated and thus reliable forecasts, as forecast uncertainty reduces with increasing sharpness (subject to calibration). To assess sharpness, we use properties of the width of prediction intervals as in Gneiting and Raftery (2007). Analog to Pasternack et al. (2018), we use the time mean intra-ensemble variance $\bar{\sigma^2}$ to asses the prediction width.

Scoring rules, like the *Continuous Ranked Probability Score* ($CRPS$), assign numerical scores to probabilistic forecasts and form attractive summary measures of predictive performance, since they address reliability and sharpness simultaneously (Gneiting et al., 2005; Gneiting and Raftery, 2007; Gneiting and Katzfusss, 2014).





Given, $F$ is the predictive cumulative distribution function (CDF) and o is the verifying observation, the CRPS is defined as

$$CRPS(F,o) = \int\limits_{-\infty}^{\infty} (F(y) - F_0(y))^2 dy, \tag{4}$$

where $F_0(y)$ is the Heaviside function and takes the values 0 or 1 if y is less than or greater equal than the observed value o. Under the assumption that the predictive CDF is a normal distribution with mean $\mu$ and variance $\sigma^2$ Gneiting et al. (2005) showed that (4) can be written as

$$CRPS(\mathcal{N}(\mu,\sigma^2),o) =$$
$$\sigma \left\{ \frac{o-\mu}{\sigma} [2\Phi\left(\frac{o-\mu}{\sigma}\right) - 1] + 2\varphi\left(\frac{o-\mu}{\sigma}\right) - \frac{1}{\sqrt{\pi}} \right\}, \tag{5}$$

where $\Phi(\cdot)$ and $\varphi(\cdot)$ denote the CDF and the PDF, respectively, of the standard normal distribution.

The CRPS is negatively oriented. A lower CRPS indicates more accurate forecasts; a CRPS of zero denotes a perfect (deterministic) forecast.

The Continuous Ranked Probability Skill Score ($CRPSS$) is, as the name implies, the corresponding skill score. A skill score relates the accuracy of the prediction system to the accuracy of a reference prediction (e.g., climatology). Thus, with a given $CRPS_F$ for the hindcast distribution and a given $CRPS_R$ for the reference distribution the $CRPSS$ can be defined as:

$$CRPSS = 1 - \frac{CRPS_F}{CRPS_R}. \tag{6}$$

Positive values of the CRPSS imply that the prediction system outperforms the reference prediction. Furthermore, this skill score is unbounded for negative values (because hindcasts can be arbitrarily bad) but bounded by 1 for a perfect forecast.

## 3 Model selection for *DeFoReSt*

We first review the decadal climate forecast recalibration strategy (*DeFoReSt*) proposed by Pasternack et al. (2018) and illustrate subsequently how a modelling strategy based on boosting and cross validation leads to an optimal selection of polynomial orders in the non-homogeneous regression model used for recalibration.

### 3.1 Review of DeFoReSt

*DeFoReSt* assumes normality for the recalibrated predictive probability distribution function (PDF) $f^{\text{Cal}}(X;t,\tau)$ for a predicted parameter $X$ for each initialization time $t \in \{1961, 1962, 1963, \ldots, 2010\}$ and lead time $\tau \in \{1, 2, 3, \ldots, 10\}$. $f^{\text{Cal}}(X;t,\tau)$ thus describes the recalibrated forecast uncertainty of a given parameter $X \sim f^{\text{Cal}}(X;t,\tau)$ or – expressed in terms of the ensemble – $f^{\text{Cal}}(X;t,\tau)$ the distribution of the recalibrated ensemble members around the recalibrated ensemble mean as a function of initialization time $t$ and forecast lead-year $\tau$. Mean $\mu_{\text{Cal}}(t,\tau)$ and variance $\sigma^2_{\text{Cal}}(t,\tau)$ of the recalibrated distribution





$f^{\text{Cal}}(X;t,\tau)$ are now modelled as linear functions of the ensemble mean $\hat{\mu}(t,\tau)$ and ensemble variance $\hat{\sigma}^2(t,\tau)$ as

$$\mu_{\text{Cal}}(t,\tau) = \alpha(t,\tau) + \beta(t,\tau)\,\hat{\mu}(t,\tau) \tag{7}$$

$$\ln(\sigma^2_{\text{Cal}}(t,\tau)) = \gamma(t,\tau)\,\hat{\sigma}^2(t,\tau). \tag{8}$$

Note, different from Pasternack et al. (2018), the logarithm in Eq. (8) ensures positiv recalibrated variance $\sigma^2_{\text{Cal}}(t,\tau)$ irrespectively of the value of $\gamma$. Hence, the recalibrated parameter $X_{\text{Cal}}$ is now distributed according to

$$X_{\text{Cal}}(t,\tau) \sim \mathcal{N}(\alpha(t,\tau) + \beta(t,\tau)\,\hat{\mu}(t,\tau), \exp(\gamma(t,\tau)\hat{\sigma}^2(t,\tau))). \tag{9}$$

$\alpha(t,\tau)$ accounts for the (unconditional) bias depending on lead year (i.e., the drift). Similarly, $\beta(t,\tau)$ accounts for the conditional bias. Thus, the expectation of the recalibrated variable $E(X_{\text{Cal}}(t,\tau)) = \alpha(t,\tau) + \beta(t,\tau)\,\hat{\mu}(t,\tau)$ can be conceived as a conditional and unconditional bias and drift adjusted ensemble mean (or "deterministic"; we call a deterministic forecast a forecast without specifying uncertainty.) forecast. Moreover, *DeFoReSt* assumes that the ensemble spread $\sigma(t,\tau)$ is sufficiently well related to the forecast uncertainty such that adequate adjustment can be realized by multiplying $\gamma(t,\tau)$.

The functional forms of $\alpha(t,\tau)$, $\beta(t,\tau)$ and $\gamma(t,\tau)$ are motivated from Gangstø et al. (2013), Kharin et al. (2012), Kruschke et al. (2015), and Sansom et al. (2016). Gangstø et al. (2013) suggested a third order polynomial in $\tau$ as a good compromise between flexibility and parameter uncertainty; the linear dependency on $t$ was used in various previous studies (Kharin et al., 2012; Kruschke et al., 2015; Sansom et al., 2016). A combination of both led to *DeFoReSt* as described in Pasternack et al. (2018):

$$\alpha(t,\tau) = \sum_{l=0}^{3}(a_{2l} + a_{(2l+1)}\,t)\,\tau^l\,, \tag{10}$$

$$\beta(t,\tau) = \sum_{m=0}^{3}(b_{2m} + b_{(2m+1)}\,t)\,\tau^m\,, \tag{11}$$

$$\gamma(t,\tau) = \sum_{n=0}^{2}(c_{2n} + c_{(2n+1)}\,t)\,\tau^n\,. \tag{12}$$

The ensemble inflation $\gamma(t,\tau)$ is, however, assumed to be quadratic at most. Pasternack et al. (2018) assumed that a higher flexibility may not be necessary.

$\alpha(t,\tau), \beta(t,\tau)$ and $\gamma(t,\tau)$ are functions of $t$ and $\tau$, linear in the parameters $a_l$, $b_m$ and $c_n$. The parameters are estimated by minimizing the average $CRPS$ over the training period following Gneiting et al. (2005) using the associated scoring function

$$\Gamma(\mathcal{N}(\alpha(t,\tau) + \beta(t,\tau)\,\hat{\mu}(t,\tau), \exp(\gamma(t,\tau)\hat{\sigma}^2(t,\tau))), o) := \overline{\text{CRPS}} =$$
$$\frac{1}{k}\sum_{j=1}^{k}\sqrt{\exp(\gamma(t,\tau)\sigma_j^2)}\left\{ Z_j\,[2\,\Phi(Z_j) - 1] + 2\,\varphi(Z_j) - \frac{1}{\sqrt{\pi}}\right\}, \tag{13}$$

where

$$Z_j = \frac{O_j - (\alpha(t,\tau) + \beta(t,\tau)\,\hat{\mu}_j(t,\tau))}{\sqrt{\exp(\gamma(t,\tau)^2\,\hat{\sigma}_j^2(t,\tau))}} \tag{14}$$





is the standardized forecast error for the $j$th forecast in the training data set. Optimization is carried out using the algorithm of
Nelder and Mead (1965) as implemented in R (R Core Team, 2016).

Initial guesses for parameters need to be carefully chosen to avoid convergence into local minima of the objective function.
Here, we obtain initial guesses for $a_l$ and $b_m$ from a standard linear model using the ensemble mean $\hat{\mu}(t,\tau)$ and polynomials of
$t$ and $\tau$ as terms in the predictor according to Eqs. (7), (10) and (11). Initial guesses for $c_0, c_1$ and $c_2$ are all zero which yields
unit inflation as $\ln(\sigma_{\text{cal}}^2(t,\tau)) = 0$ leads to $\sigma_{\text{cal}}^2(t,\tau) = 1$. Convergence to the global minimum is facilitated, however, cannot
be guaranteed.

An alternative to minimization of the CRPS is maximization of the likelihood. Here, CRPS grows linearly in the prediction
error, in contrast to the likelihood which grows quadratically (Gneiting et al., 2005). Thus a maximization of the likelihood is
more sensitive to outliers and extreme events (Weigend and Shi, 2000; Gneiting and Raftery, 2007). This implies a prediction
recalibrated using likelihood maximization is more likely to be underconfident than a prediction recalibrated using CRPS
minimization (Gneiting et al., 2005).

We use cross-validation with a 10-year moving validation period as proposed by Pasternack et al. (2018) to ensure fair con-
ditions for assessing the benefit of *DeFoReSt*. This means, the parameters $a_l, b_m$ and $c_n$ needed for recalibrating one hindcast
experiment with 10 lead years (e.g. initialization 1963, forecasting years 1964 to 1973) are estimated via those hindcasts which
are initialized outside that period (e.g. here hindcasts initialized 1962; 1974; 1975,...). This procedure is repeated for every
initialization year $z \in \{1960, 1961, 1962, \ldots, 2010\}$. Figure 1 shows an illustration of this setting.

### 3.2 *Boosted recalibration* and cross-validation

In Eq. 8, we followed Pasternack et al. (2018) with a multiplicative term $\gamma(t,\tau)$ to adjust the spread. From now on, we follow
the suggestion and notation from Messner et al. (2017) and include an additive term ($\gamma(t,\tau)$) and multiplicative term ($\delta(t,\tau)$).
The model for the calibrated ensemble variance (Eq. (8)) changes to

$$\ln(\sigma_{\text{Cal,boost}}^2(t,\tau)) = \gamma(t,\tau) + \delta(t,\tau)\hat{\sigma}^2(t,\tau). \tag{15}$$

Note the change in definition for $\gamma(t,\tau)$!

$\alpha(t,\tau)$, $\beta(t,\tau)$, $\gamma(t,\tau)$ and $\delta(t,\tau)$ are modelled using a similar approach as in Eqs. 10–12 where we now use orthogonalized
polynomials to address for the lead time dependency of these corrections terms. In light of a model selection, this has the
advantage that the individual predictors are now uncorrelated. Moreover, we now use orthogonalized polynomials of order 6
in lead time $\tau$, assuming that this is sufficiently large to capture all features of lead time dependent drift ($\alpha(t,\tau)$), conditional



bias ($\beta(t,\tau)$) and ensemble dispersion ($\gamma(t,\tau)$ and $\delta(t,\tau)$); the dependence on initialization time $t$ is kept linear:

$$\alpha(t,\tau) = \sum_{l=0}^{6}(a_{2l} + a_{(2l+1)}t)P_l(\tau), \tag{16}$$

$$\beta(t,\tau) = \sum_{m=0}^{6}(b_{2m} + b_{(2m+1)}t)P_m(\tau), \tag{17}$$

$$\gamma(t,\tau) = \sum_{n=0}^{6}(c_{2n} + c_{(2n+1)}t)P_n(\tau), \tag{18}$$

200  $$\delta(t,\tau) = \sum_{p=0}^{6}(d_{2p} + d_{(2p+1)}t)P_p(\tau). \tag{19}$$

Here, $P_l(\tau), P_m(\tau), P_n(\tau)$ and $P_p(\tau)$ are orthogonalized polynomials of order $l, m, n$ and $p$, which are provided by the R-function `poly`.

We apply boosting for non-homogeneous regression problems as proposed by Messner et al. (2017) for estimating $a_l$, $b_m$, $c_n$ and $d_p$. The algorithm iteratively seeks the minimum of a loss function (negative log-likelihood or CRPS) by identifying and

205  updating only the most relevant terms in the predictor. This is realized with the R-package `crch` for non-homogeneous boosting (Messner et al., 2017, available from http://cran.r-project.org/) which uses a minimization of the negative log-likelihood by default instead of minimizing the CRPS. Judging from our experience, for the problem at hand, the difference in using one or the other loss functions appears to be small. The above mentioned effect of outliers and extremes on dispersivity described by Gneiting et al. (2005) should be rather small here, since annual aggregated values are recalibrated.

210  In each iteration, the negative partial derivatives

$$r = -\frac{\partial l(\mu,\sigma)}{\partial\mu}; \quad s = -\frac{\partial l(\mu,\sigma)}{\partial\sigma}, \tag{20}$$

of the negative log-likelihood for a single observation $y$

$$l(\alpha + \beta\mu, \gamma + \delta\sigma; y) = -\log\left(\frac{1}{\gamma + \delta\sigma}\phi\left(\frac{y - \alpha + \beta\mu}{\gamma + \delta\sigma}\right)\right), \tag{21}$$

is obtained. Where $\phi$ is the probability density function of the normal distribution, $\mu$ the ensemble mean and $\sigma$ the ensemble

215  standard deviation corresponding to the initialization time $t$ and lead time $\tau$ of the observation $y$. Pearsons correlation coefficient between each predictor term (e.g., $t$ or $t\tau^2$) and the partial derivatives $r$ and $s$ (Eq. (20)) estimated over every available $t \in \{1961, 1962, 1963, \ldots, 2010\}$ and $\tau \in \{1, 2, 3, \ldots, 10\}$ is used to identify and update the most influential term in the predictor. The parameter associated to the term with the highest correlation is updated by their correlation coefficient multiplied with a predefined stepsize $\nu$. Schmid and Hothorn (2008) showed that the choice of $\nu$ is only of minor importance and suggested a

220  value of 0.1. Nonetheless, from personal experience we use $\nu = 0.05$, which turns out to be more appropriate for our study. A distinct feature of boosting for non-homogeneous regression is, that *both* mean and standard deviation of a forecast distribution are taken into account, but for each iteration step only *one* parameter (either associated to the mean $\mu_{\mathrm{Cal,boost}}$ or variance $\sigma_{\mathrm{Cal,boost}}$) is updated: the one leading to the largest improvement of the objective function. Only those parameters associated



to the most relevant predictor terms are updated; parameters of less relevant terms remain zero. The algorithm is described in more detail in Messner et al. (2017).

A cross-validation (CV) approach is used to identify the iteration with the set of parameter estimates with maximum predictive performance. Currently, CV is carried out after each boosting iteration. The data is split into 5 parts, each part consist of approx. 10 years in order to reflect conditions of decadal prediction. For each part, a recalibrated prediction is computed, with the model trained on the remaining 4 parts. The full negative log-likelihood results from summing Eq. (21) for all available $t$ and $\tau$ and the associated observations $y$. The iteration step with minimum negative log-likelihood is considered best. We allow a maximum number of 500 iterations.

Analog to standard *DeFoReSt*, the previously described modelling procedure (boosting and CV for iteration selection) is carried out in a cross-validation setting (second level of CV) for model validation. A 10-year moving validation period (see Sec. 3.1) leads to cross-validation. For example, to recalibrate the hindcast initialized 1963 including lead years 1964 to 1973, all hindcasts which are *not* initialized within that period (e.g. $t \in \{1960, 1974, 1975, 1976, \ldots, 2010\}$) are used for boosting *DeFoReSt*.

## 4 Calibrating toy model experiments

To assess the model selection approach for *DeFoReSt* we consider two extreme toy model experiments to generate pseudo-forecasts, as introduced by Pasternack et al. (2018). They are designed as follows

1. the predictable signal is *stronger* than the unpredictable noise,

2. the predictable signal is *weaker* than the unpredictable noise.

These experiments are controlled by five further parameters:

$\eta$ determines the ratio between the variance of the predictable signal and the variance of the unpredictable noise, it controls potential predictability, see Pasternack et al. (2018). We investigate two cases: $\eta = 0.2$ (low potential predictability) and $\eta = 0.8$ (high potential predictability).

$\chi(t, \tau)$ specifies the unconditional bias added to the predictable signal,

$\psi(t, \tau)$ specifies analogously the conditional bias, and

$\omega(t, \tau)$ specifies the conditional dispersion of the forecast ensemble.

$\xi(t, \tau)$ controls analogously the unconditional dispersion and has not been used in Pasternack et al. (2018).

The coefficients for Bias (drift), conditional bias and effects in the ensemble dispersion are calculated by calibrating *Prototype* surface temperature data with *HadCrut4* observations. Thus $\chi(t, \tau), \psi(t, \tau), \omega(t, \tau)$ and $\xi(t, \tau)$ based on the same polynomial structure as used for the calibration parameters $\alpha(t, \tau), \beta(t, \tau), \gamma(t, \tau)$ and $\delta(t, \tau)$ (see (16)-(19)) (a detailed description of the





toy model design is given in Appendix A). In the following, when we discuss the polynomial lead time dependency of the toy models systematic errors we refer to the polynomial order of $\alpha(t,\tau)$, $\beta(t,\tau)$, $\gamma(t,\tau)$ and $\delta(t,\tau)$. Note that the corresponding polynomials are also orthogonalized as in (16) -(19).


For an assessment of the model selection approach, we are using seven different toy-model setups per value of $\eta$. Each setup uses different orders of polynomial lead time dependency for imposing the above mentioned systematic deviations on the predictable signal. One toy model setup is designed such that the corresponding systematic deviations could be perfectly addressed by *DeFoReSt*. Additionally, there are other setups with systematic deviations based on a lower/higher polynomial

order than what is used for *DeFoReSt*. Thus we compare pseudo-forecasts from setups which require model structures for recalibration given in Tab. 1.

| Setup | $\alpha(t,\tau) =$ | $\beta(t,\tau) =$ | $\gamma(t,\tau) =$ | $\delta(t,\tau) =$ |
|---|---|---|---|---|
| | $(a_0 + a_1 t)P_0(\tau) + ...$ | $(b_0 + b_1 t)P_0(\tau) + ...$ | $(c_0 + c_1 t)P_0(\tau) + ...$ | $(d_0 + d_1 t)P_0(\tau) + ...$ |
| 1 | $(a_2 + a_3 t)P_1(\tau)$ | $(b_2 + b_3 t)P_1(\tau)$ | $(c_2 + c_3 t)P_1(\tau)$ | $(d_2 + d_3 t)P_1(\tau)$ |
| 2 | $(a_4 + a_5 t)P_2(\tau)$ | $(b_4 + b_5 t)P_2(\tau)$ | $(c_4 + c_5 t)P_2(\tau)$ | $(d_4 + d_5 t)P_2(\tau)$ |
| 3 | $(a_6 + a_7 t)P_3(\tau)$ | $(b_6 + b_7 t)P_3(\tau)$ | $(c_6 + c_7 t)P_3(\tau)$ | $(d_6 + d_7 t)P_3(\tau)$ |
| DeFoReSt | $\sum_{l=1}^{3}(a_{2l} + a_{(2l+1)}t)P_l(\tau)$ | $\sum_{m=1}^{3}(b_{2m} + b_{(2m+1)}t)P_m(\tau)$ | $\gamma(t,\tau) = 0$ | $\sum_{p=1}^{2}(d_{2p} + d_{(2p+1)}t)P_p(\tau)$ |
| 4 | $(a_8 + a_9 t)P_4(\tau)$ | $(b_8 + b_9 t)P_4(\tau)$ | $(c_8 + c_9 t)P_4(\tau)$ | $(d_8 + d_9 t)P_4(\tau)$ |
| 5 | $(a_{10} + a_{11} t)P_5(\tau)$ | $(b_{10} + b_{11} t)P_5(\tau)$ | $(c_{10} + c_{11} t)P_5(\tau)$ | $(d_{10} + d_{11} t)P_5(\tau)$ |
| 6 | $(a_{12} + a_{13} t)P_6(\tau)$ | $(b_{12} + b_{13} t)P_6(\tau)$ | $(c_{12} + c_{13} t)P_6(\tau)$ | $(d_{12} + d_{13} t)P_6(\tau)$ |
| | unconditional | conditional | unconditional | conditional |
| | bias | bias | dispersion | dispersion |

**Table 1.** Overview of the different toy model setups and the corresponding polynomial lead time dependencies.

As mentioned before, the functions $\chi(t,\tau)$, $\psi(t,\tau)$, $\xi(t,\tau)$ and $\omega(t,\tau)$ in the toy model experiments are based on the parameters estimated for calibrating the *MiKlip Prototype* ensemble global mean surface temperature against *HadCRUT4* observations. Here, $\chi(t,\tau)$, $\psi(t,\tau)$, $\xi(t,\tau)$ and $\omega(t,\tau)$ are based on ratios of polynomials up to $3^{rd}$ order w.r.t. lead time.

Based on our experience we assume that systematic errors with higher than $3^{rd}$ order polynomials could not be detected sufficiently well within the MiKlip Prototype experiments. Therefore, the coefficients for the $4^{th}$ to $6^{th}$ order polynomials are deduced from the coefficient magnitude of the $1^{st}$ to $3^{rd}$ order polynomial. Here, Fig. 2 shows the coefficients which were obtained from calibrating the MiKlip Prototype global mean surface temperature with cross-validation (see Pasternack et al. (2018)), assuming a $3^{r}d$ order polynomial dependency in lead years for $\alpha(t,\tau)$, $\beta(t,\tau)$, $\gamma(t,\tau)$ and $\delta(t,\tau)$. Those coefficients

associated with terms describing the lead time dependence exhibit roughly the same order of magnitude. Thus, we assume the coefficients associated to $4^{th}$ to $6^{th}$ order polynomials being of the same order of magnitude. An overview of the applied coefficient values is given in Appendix A.





Analogously to the MiKlip experiment, the toy model uses 50 start years, each with 10 lead years, and 15 ensemble members. The corresponding pseudo-observations run over a period of 59 years in order to cover lead year 10 of start year 50.

For each toy model setup we calculated the *Ensemble Spread Score ESS*, the *Mean Squared Error MSE*, time mean intra-ensemble variance and the *Continuous Ranked Probability Skill Score CRPSS* of pseudo-forecasts recalibrated with boosting. Reference for the skill-score are forecasts recalibrated with *DeFoReSt*. All scores have been calculated using cross-validation with an annually moving calibration window with a width of 10 years (see Pasternack et al. (2018)).

To ensure a certain consistency 100 pseudo-forecasts are generated from the toy model and evaluated as described above. The
scores presented are all mean values over these 100 experiments. In particular, to assess a significant improvement of *boosted recalibration* over *DeFoReSt* w.r.t. $CRPSS$ the 2.5% and 97.5% percentiles are also estimated from this 100 experiments.

### 4.1     Toy model with high potential predictability ($\eta = 0.8$)

Figures 3a-c show the $MSE$ for 7 different setups (see 4). Panel 3a shows the result without any post-processing (raw pseudo-forecasts), panel 3b with *DeFoReSt* and panel 3c with *boosted recalibration*. Here, the performance of both post processing
methods is strongly superior to the raw pseudo-forecast output. As *DeFoReSt* uses third order polynomials in lead time to capture conditional and unconditional biases, it performs equally well as the *boosted calibration* for the first four setups; for setups using higher order polynomials *boosted calibration* is superior.

Regarding the $ESS$ (Figs. 4a-c) shows that the raw pseudo-forecasts are widely fluctuating between under- and overdispersiveness ($ESS$-values from 0.1 to 1.7), depending on the corresponding complexity of the imposed systematic errors (different
setups). Corresponding to this the post processed pseudo-forecasts are more reliable with $ESS$-values close to 1. The *boosted recalibration* approach is superior to the recalibration with *DeFoReSt* for every lead year. The improvement is largest for setups 4-6, because *DeFoReSt* is limited to third order polynomials and cannot account for higher polynomial orders of these setups.

The post-processing methods are further compared by calculating the time mean intra-ensemble variance (see Figs. 5a-c). For every setup the intra-ensemble variance of the raw pseudo-forecasts is higher than the intra-ensemble variance of corresponding
post-processed forecasts. Comparing *DeFoReSt* with the boosted recalibration reveals that the sharpness of the first approach is larger for setups 1 to 3 and the 'DeFoReSt setup', leading particularly for the first 3 setups to an overconfidence (see 4b). However for setups 4 to 6 *DeFoReSt* exhibits a smaller sharpness, which still results in combination with the increased $MSE$ (see 3b) to underdispersiveness.

A joint measure for sharpness and reliability is the $CRPS$ and its skill-score, the $CRPSS$. Figure 6 shows the $CRPSS$
of the different pseudo-forecasts with *boosted recalibration*, where pseudo-forecasts recalibrated with *DeFoReSt* are used as reference, i.e. positive values imply that *boosted recalibration* is superior to *DeFoReSt*. Colored dots in Fig. 6 denote significance in the sense that the 0.025 and 0.975 quantiles from the 100 experiments do not include 0. Regarding setups 1 to 3 and the 'DeFoReSt setup', the $CRPSS$ is neither significantly positive nor negative for all lead years, except lead year 1 of setup 3. On the other hand, for setups 4 to 6 the *boosted recalibration* outperforms the recalibration with *DeFoReSt* with values
of the $CRPSS$ between 0.1 and 0.4. Again, this is likely due to *DeFoReSt* assuming third order polynomials in lead time to capture conditional and unconditional biases, second order for dispersion and therefore does not account for systemetic errors





based on higher orders. However, Fig. 6 suggegsts that *boosted recalibration* can account for systematic errors with various levels of complexities.

## 4.2   Toy model with low potential predictability ($\eta = 0.2$)

Figures 7a-c show the $MSE$ of the different pseudo-forecasts for a toy model setup with a low potential predictability. One can see that both post processing approaches lead to a strong improvement compared to the raw pseudo-forecasts; both approaches work roughly equally well for all setups. Compared to the previous section ($\eta = 0.8$), the $MSE$ of the pseudo-forecasts has increased due to a smaller signal-to-noise-ratio.

Regarding the $ESS$ (see Fig. 8a-c), reveals that compared to the pseudo-forecasts with high predictability the raw simula-
tions from different toy models are underdispersive for almost all lead years ($ESS$-values smaller than 1). The pseudo-forecasts show again an increased reliability after recalibration, with $ESS$-values close to 1. For every lead year, *boosted recalibration* is superior to *DeFoReSt*; the latter leads to slightly overconfident recalibrated forecasts.

Figures 9a-c show the time mean intra-ensemble variance of the raw and recalibrated pseudo-forecasts. For every setup the intra-ensemble variance of the different pseudo-forecasts has decreased due to recalibration (with and without boosting).
Comparing *DeFoReSt* with *boosted recalibration* reveals a smaller intra-ensemble variance for every setup, leading to an overconfidence for every lead year as observed in Fig 8b.

In the low potential predictability setting ($\eta = 0.2$) the ensemble variance is larger as the total variance in the toy model is constrained to one. Thus reducing $\eta$ leads to an increase in ensemble spread.

Figure 10 shows the $CRPSS$ of the pseudo-forecasts with *boosted recalibration* with *DeFoReSt* as reference. The low
potential predictability leads to a reduced $CRPSS$ compared to the setting with $\eta = 0.8$. The improoevement due to *boosted recalibration* is also smaller. Only the first two lead years of setups 4-6 are significantly different from zero. This suggests that the improvement due to *boosted recalibration* decreases with a decreasing potential predictability of the forecasts.

## 5   Calibrating decadal climate surface temperature forecasts

While in Sec. 4 *DeFoReSt* and *boosted recalibration* were compared by the use of different toy model data, in this section these
two approaches will be applied to surface temperature of MiKlip Prototype runs with MPI-ESM-LR. Here, global mean and spatial mean values over the North Atlantic subpolar gyre (60°-10°W, 50°-65°N) region will be analyzed.

Here, we discuss which predictors are identified by *boosted recalibration* as most relevant and we compute the $ESS$, the $MSE$ the intra-ensemble variance and the $CRPSS$ with respect to climatology for both recalibration approaches. The scores have been calculated for a period from 1960 to 2010. In this section, a 95% confidence interval was additionally calculated
for these metrics using a bootstrapping approach with 1000 replicates. For bootstrapping we draw a new pair of dummy time series with replacement from the original validation period and calculate these scores again. This procedure has been repeated 1000 times. Furthermore, all scores have been calculated using cross-validation with a yearly moving calibration window with a width of 10 years (see Sec-3.1)





## 5.1 Global mean surface temperature

Figure 11 shows the coefficients estimated by *boosted recalibration* for global mean surface temperature. The predictors are standardized, i.e. larger coefficients imply larger relevance of the corresponding predictors for the recalibration. Model selection is based on negative log-likelihood minimization in a cross-validation setup, as proposed by Pasternack et al. (2018). Thus for every training period different coefficients are obtained. The resulting distributions are represented in a box-and-whisker-plot, which also allows an assessment of the variability in coefficient estimates.

Most relevant are the coefficients $a_0$ and $a_1$, associated with unconditional bias ($a_0$) and the linear dependence on the start year ($a_1$). This is followed by $b_0$ in the conditional bias. In general, coefficients associated with first and second order terms in the lead time dependence ($a_2$, $a_4$, $b_2$, $b_4$) are dominating. Those coefficients describing the interaction between linear start year and first or second order lead year dependency (e.g., $a_3$, $b_3$, $c_3$, $b_5$, $c_5$) have also some impact.

The recalibration of ensemble dispersion is mostly influenced by a linear start year dependence in the unconditional term 350 ($c_1$) and in the conditional term $d_0$. Higher terms are of minor relevance.

The performance of the ensemble mean of the raw forecast (black), recalibrated with *DeFoReSt* (blue) and with *boosted recalibration* is measured with the $MSE$ shown in Fig. 12a. While a strong drift (lead-year dependence) influences the MSE for the raw forecasts, both recalibrated variants exhibit a smaller and roughly constant $MSE$ across all $\tau$. This decrease in $MSE$ is a result of adjusting the unconditional and conditional bias ($\alpha(t,\tau)$ and $\beta(t,\tau)$).

Figure 12b evaluates the ensemble spread and shows the $ESS$. The raw pseudo-forecast is underdispersive (ESS< 1) for all lead years and needs recalibration. The recalibrated forecasts show an adequate ensemble spread in both cases (ESS close to 1) for all lead years. *Boosted recalibration* (red) outperforms *DeFoReSt* which becomes slightly under-/overdispersive for the first/last lead years. However, the differences in ESS between *boosted recalibration* and *DeFoReSt* are not significant.

Figure 12c shows the intra-ensemble variance (temporal average) across lead-years $\tau$. The ensemble variances of the raw 360 forecast and *DeFoReSt* are roughly equal, while *boosted recalibration* adjust the ensemble variance.

Compared to raw and *DeFoReSt*, the intra-ensemble variance of *boosted recalibration* is larger for lead year 1 and smaller for lead years 3 to 10. *Boosted recalibration* is sufficiently flexible to adjust the ensemble variance to a value close to the MSE. This consistent behaviour is roughly constant over lead years.

Although, *boosted recalibration* shows mostly a smaller ensemble variance (lead years 3-10) than *DeFoReSt*, both recalibra-365 tion approaches are roughly equal when the performance is assessed with the $CRPSS$ with climatological reference (Fig. 12d). Thus, the different time mean intra-ensemble variances resulting from recalibration with and without boosting have a minor impact on the $CRPSS$.

Here, the $CRPSS$ of both models is around 0.8 for all lead years w.r.t. climatological forecast. In contrast, the raw forecast is inferior to the climatological forecast for most lead years, except lead years 3-6, where the raw forecast has positive skill, 370 which could be attributed to the fact that temperature anomalies are considered. This implies that the observations and the raw forecast have the same mean value 0. This mean value seems to be crossed by the raw forecast mainly between lead 4 and 5.





## 5.2 North Atlantic mean surface temperature

Figure 13 shows the coefficients of the corresponding standardized predictors which were estimated using *boosted recalibration* for North Atlantic surface temperature. Analogously to the global mean surface temperature, model selection is used within a cross-validation setup and the resulting coefficient distributions are shown in a box-and-whisker-plot. Here, the terms for the unconditional ($a_i$) and conditional bias ($b_j$) for the linear start year dependency ($a_i t$ and $b_j t$) and the first polynomial order lead time dependency ($a_i P_1(\tau)$ and $b_j P_1(\tau)$) are most relevant. Moreover, the linear interaction between lead time and initialization time ($a_3 t P_1(\tau)$) was identified as a relevant factor for the unconditional bias. Regarding the coefficients corresponding to the unconditional ($c_k$) and conditional ($d_l$) ensemble dispersion, one can see that the linear start and lead year dependencies ($c_1 t$, $c_2 P_1(\tau)$ and $d_1 t$, $d_2 P_1(\tau)$), as well as the interaction ($d_3 t P_1(\tau)$) between these two coefficients have the most impact.

Figure 14a shows the $MSE$ of the raw forecast (black), *DeFoReSt* and *boosted recalibration*, where both recalibrated forecasts perform roughly equal. The raw forecast is inferior to both post processed forecast, mostly due to missing correction of unconditional and conditional biases. Compared to global mean temperature (Fig. 12a), $MSE$ for the North Atlantic temperature is generally larger. Thus potential predictability for the North Atlantic surface temperature is smaller than in the global case.

Regarding the reliability both recalibrated forecasts show also an $ESS$ close to one for all lead years for the North Atlantic surface temperature (Fig. 14b), which is similar to the outcome of the global mean temperature (Fig. 12b). Again *bossted recalibration* outperforms *DeFoReSt*, the latter becomes slightly underdispersive for later lead years. However, the differences in $ESS$ for both recalibration approaches are not significant. The raw forecast's reliability is obviously inferior here, as it is significantly underdispersive for lead years 1 to 3 and overdispersive for lead years 5 to 6.

The mentioned lower potential predictability for the North Atlantic manifests also in a 10-times larger ensemble variance, cf. Fig. 14c. Noteworthy is here, that due to the smaller potential predictability in this region, the ensemble variance of both recalibrated forecasts is similar across the lead time and different from the raw forecast. A lower predictability of the North Atlantic surface temperature yields also a smaller $CRPSS$ w.r.t. climatology for both recalibrated forecasts, Fig. 14d. Again, both recalibrated forecasts perform roughly equal for all lead years and are also significantly to the raw forecast.

## 6 Conclusions

Pasternack et al. (2018) proposed the recalibration strategy for decadal prediction (*DeFoReSt*) which adjusts non-homogeneous regression (Gneiting et al., 2005) to problems of decadal predictions. Characteristic problems here are a lead time and initialization time dependency of unconditional, conditional biases and ensemble dispersion. *DeFoReSt* assumes third order polynomials in lead time to capture conditional and unconditional biases, second order for dispersion, first order for initialization time dependency. Although, Pasternack et al. (2018) shows that *DeFoReSt* leads to improvement of ensemble mean and probabilistic decadal predictions, it is not clear whether these polynomials with predefined orders are optimal. This calls for a model selection approach to obtain a recalibration model as simple as possible and as complex as needed. We thus propose here not to





restrict orders a priori to such a low order but use a systematic model selection strategy to determine optimal model orders. We use the non-homogeneous boosting strategy proposed by Messner et al. (2017) to identify the most relevant terms for recalibration. The recalibration approach with boosting (called *boosted recalibration*) starts with order six polynomials in lead time and first order in initialization time to account for the unconditional and conditional bias, as well as for ensemble dispersion.

Besides other common parameter estimation and model selection approaches like stepwise regression or LASSO (Tibshi-

rani, 1996), which are designed for predictions of mean values, non-homogeneous boosting adjusts mean and variance, it automatically selects the most relevant input terms for post-processing ensemble predictions with non-homogeneous (i.e. varying variance) regression. Boosting iteratively seeks the minimum of a loss function (here the log-likelihood) and updates only the one coefficient with the largest improvement of the fit; if the iteration is stopped before a convergence criterion is fulfilled those coefficients not considered until then are kept at zero. Thus, boosting is able to handle statistical models with a large

number of variables.

We investigated *boosted recalibration* using toy model simulations with high ($\eta = 0.8$) and low potential predictability ($\eta = 0.2$) and errors with different complexities in terms of polynomial orders in lead time were imposed. *Boosted recalibration* is compared to *DeFoReSt*. The $CRPSS$, the $ESS$, the time mean intra-ensemble variance (a measure for sharpness) and the $MSE$ assess the performance of the recalibration approaches. Scores are calculated with 10 year block-wise cross-validation

(Pasternack et al., 2018) and with 100 pseudo-forecasts for each toy model simulation.

Irrespective of the complexity of systematic errors and the potential predictability, both recalibration approaches lead to an improved reliability with $ESS$ close to one. Sharpness and $MSE$ can also be improved with both recalibration approaches. Given a high potential predictability ($\eta = 0.8$), *boosted recalibration* is superior to *DeFoReSt* if systematic errors are less complex than a $3^r d$ order polynomial in lead time, implied by the $CRPSS$ of the pseudo-forecasts recalibrated with *boosted*

*recalibration* and *DeFoReSt* as reference. Moreover, a significant improvement for almost all lead years can be observed if the complexity of systematic errors is larger than 3rd order order polynomials in lead time. The gain w.r.t. *DeFoReSt* can hardly be observed for a low potential predictability ($\eta = 0.2$), as the $CRPSS$ shows only for two lead years a significant improvement for the above mentioned complexities. This is due to a generally weaker predictable signal, and thus a weaker impact of systematic error terms in higher order of the polynomial. The improvement due boosting increases with the imposed

predictability. However, the presented toy model experiments suggest the use of *boosted recalibration* due to higher flexibility without loss of skill.

Analogously to Pasternack et al. (2018), we recalibrated mean surface temperature of the MiKlip Prototype decadal climate forecasts, spatially averaged over the North Atlantic subpolar gyre region and a global mean. Pronounced predictability for these cases has been identified by previous studies (e.g., Pohlmann et al., 2009; van Oldenborgh et al., 2010; Matei et al.,

2012; Mueller et al., 2012). Nonetheless, both regions are also affected by a strong model drift (Kröger et al., 2018). For the global mean surface temperature, we could identify the linear start year dependency of the unconditional bias as a major factor. Moreover, it turns out that polynomials of lead year dependencies with order greater than 2 are of minor relevance.



Regarding the probabilistic forecast skill ($CRPSS$), *DeFoReSt* and *boosted recalibration* perform roughly equal, implying that the polynomial structure of *DeFoReSt*, chosen originally from personal experience, turns out to be quite appropriate. Both

recalibration approaches are reliable and outperforming the climatological forecast with a $CRPSS$ near $0.8$.

For the North Atlantic region, the linear start year and lead year dependencies of the unconditional and conditional biases show the largest relevance; also the linear interaction between lead time and initialization time of the unconditional bias has a certain impact. The coefficients corresponding to the unconditional and conditional ensemble dispersion, show a minor relevance compared to the errors related to the ensemble mean.

Also for the North Atlantic surface temperature both post-processing approaches are performing roughly equal; they are reliable and superior to climatology w.r.t. $CRPSS$. However, the $CRPSS$ for the North Atlantic case is generally smaller than for the global mean.

This study shows that *boosted recalibration*, i.e. recalibration model selection with nonhomogeneous boosting allows a parametric decadal recalibration strategy with an increased flexibility to account for lead time dependent systematic errors.

However, while we increased the polynomial order to capture lead time dependent features, we still assumed a linear dependency in initialization time. As this model selection approach reduces parameters by eliminating irrelevant terms, this opens up the possibility to increase flexibility (polynomial orders) also in terms related to the start year.

Based on simulations from a toy model and the MiKlip decadal climate forecast system we could demonstrate the benefit of model selection with boosting (*boosted recalibration*) for recalibrating decadal predictions, as it decreases the number of

predictors without being inferior to the state-of-the-art recalibration approach *DeFoReSt*.

*Code and data availability.* The *HadCRUT4* global temperature data set used in this study is freely accessible through the Climatic Research Unit at the University of East Anglia (http://www.cru.uea.ac.uk). The MiKlip Prototype data used for this paper are from the BMBF-funded project MiKlip and are available on request. The post-processing, toy model and cross-validation algorithms are implemented using GNU licensed free software from the R Project for Statistical Computing (http://www.r-project.org) and can be found under

https://doi.org/10.5281/zenodo.3975758 (Pasternack et al., 2020).



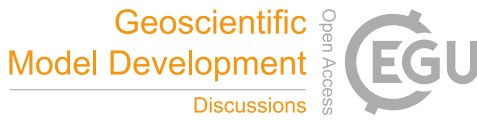

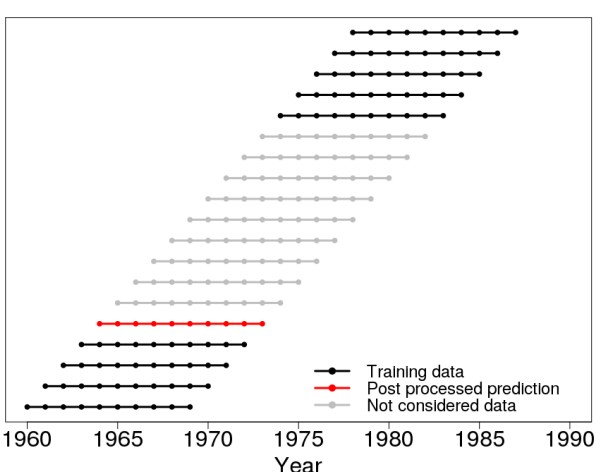

**Figure 1.** Schematic overview of the cross-validation setting for a decadal climate prediction, initialized in 1964 (red dotted line). All hindcasts which are initialized outside the prediction period are used as training data (black dotted lines). A hindcast which is initialized inside the prediction period is not used for training (gray dotted lines).



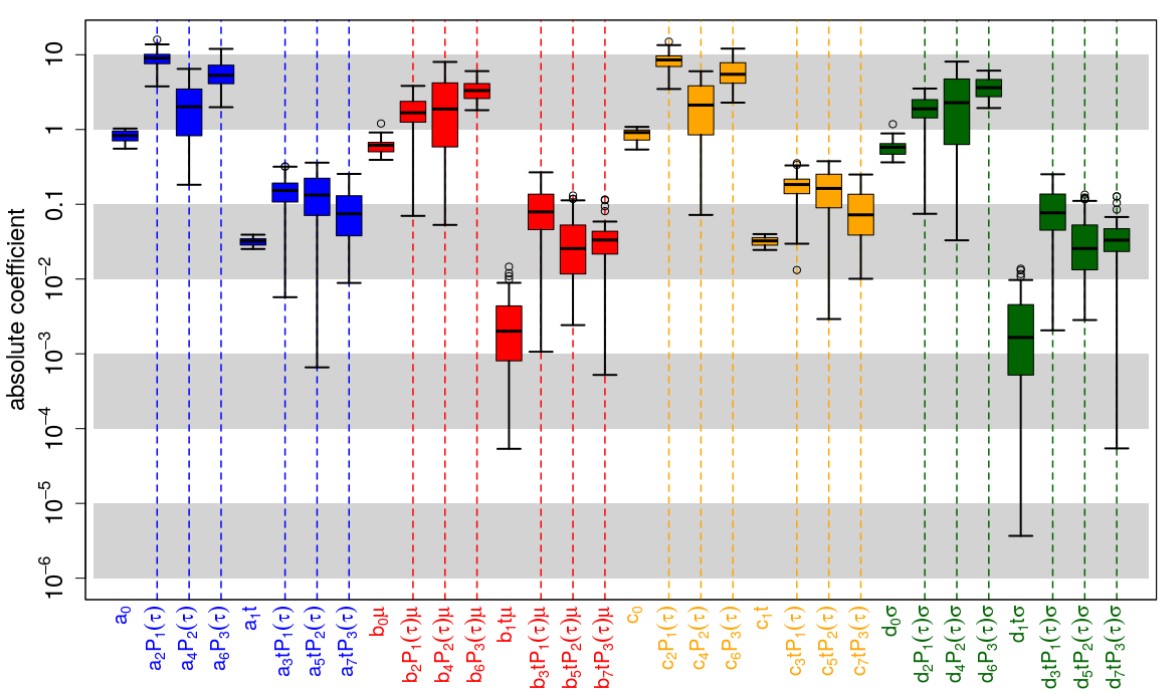

**Figure 2.** Coefficient estimates for recalibrating global mean 2m-Temperature of the *MiKlip Prototype System* with a third order polynomial lead time time dependency for the unconditional and conditional bias ans dispersion. Here, non-homogeneous boosting is not applied and all polynomials are orthogonalized, i.e. $P_1(\tau), P_2(\tau), P_3(\tau)$ refers to the order of the corresponding polynomial. Colored boxes represent the inter-quartile range $(IQR)$ around the median (central, bold and black line) for coefficient estimates from the cross-validation setup; Whiskers denote maximum 1.5IQR. Coefficients are grouped according to correcting unconditional bias (blue), conditional bias (red), unconditional dispersion (orange) and conditional dispersion (green). Values refer to coefficients $a_0, b_0, c_0, d_0, ..., a_6, b_6, c_6, d_6$ and not to the product between these coefficients and the corresponding predictors (e.g. $a_2 P_1(\tau)$ refers to $a_2$). Vertical dashed bars highlight coefficients related to lead time dependent terms.





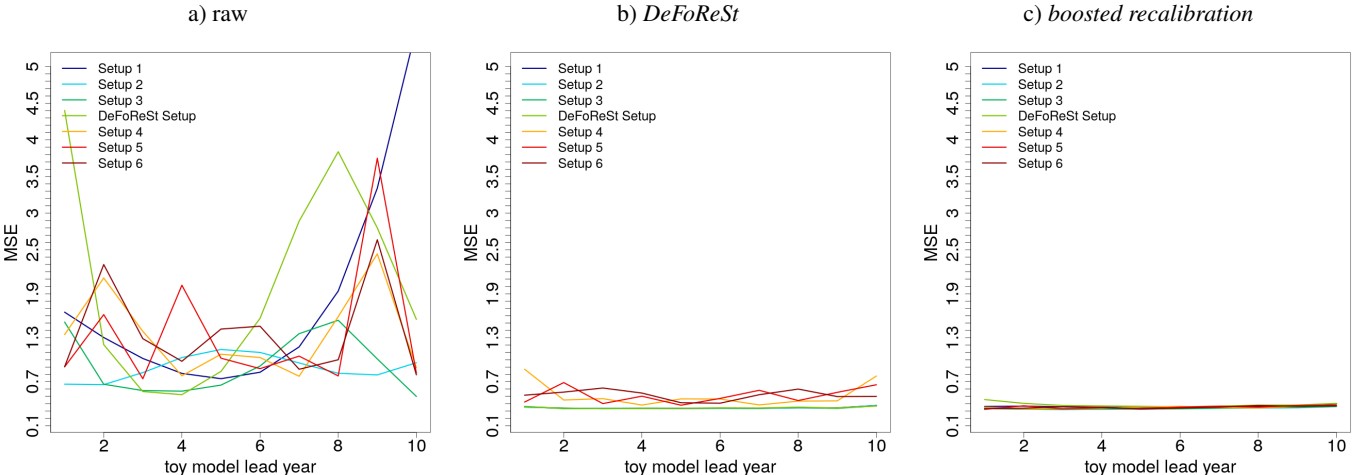

**Figure 3.** Mean squared error ($MSE$) of different toy model setups with high potential predictability ($\eta = 0.8$, colored lines). a) raw pseudo-forecast, b) post-processing with *DeFoReSt* and c) post-processing with *boosted recalibration*.





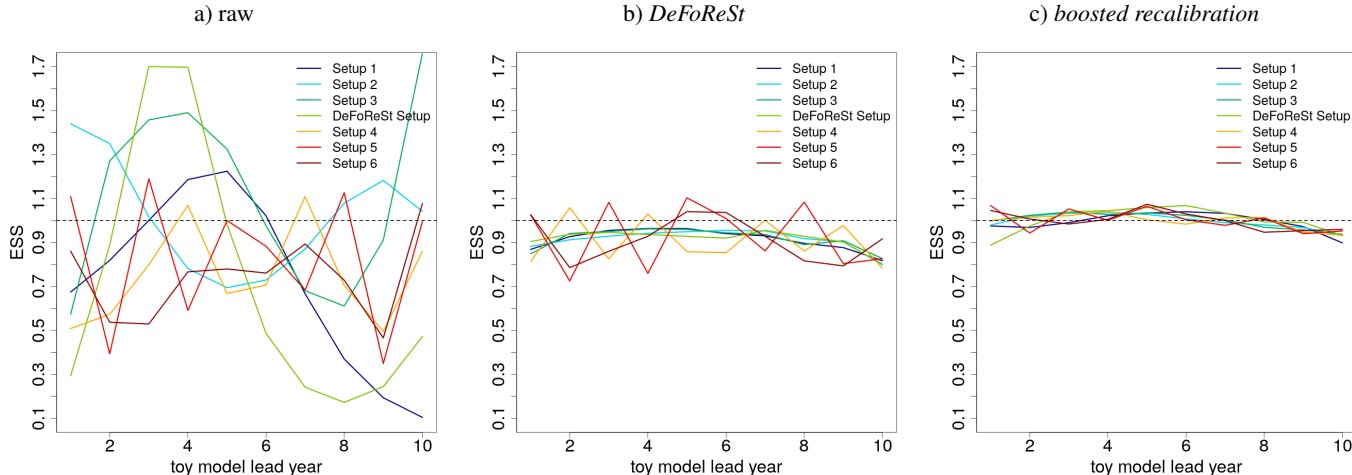

**Figure 4.** Ensemble spread score ($ESS$) of different toy model setups with high potential predictability ($\eta = 0.8$, colored lines). a) raw pseudo-forecast, b) post-processing with *DeFoReSt* and c) post-processing with *boosted recalibration*.





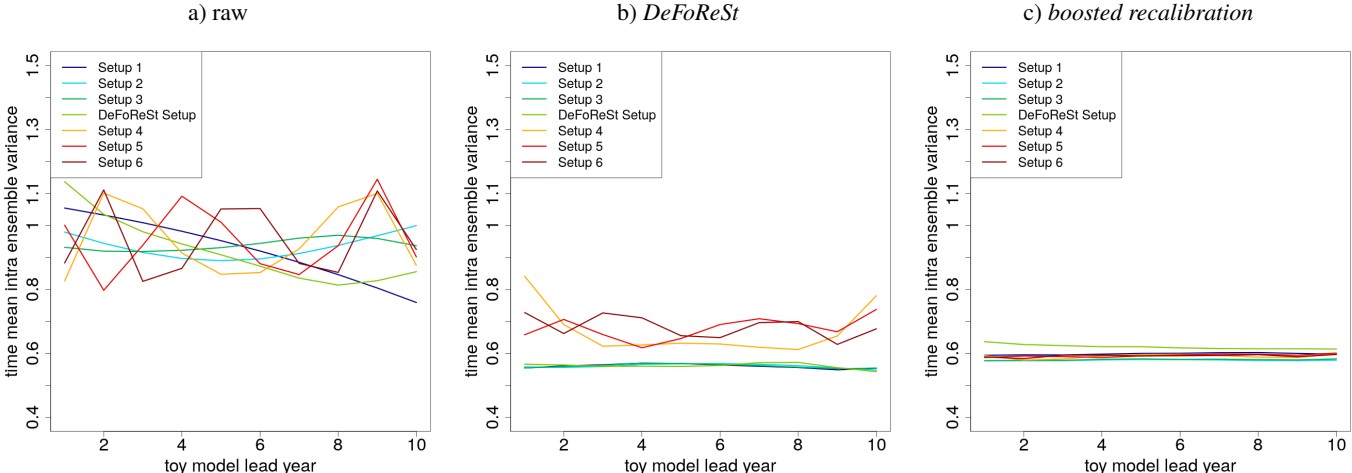

**Figure 5.** Intra-ensemble variance (temporal average) of different toy model setups with high potential predictability ($\eta = 0.8$, colored lines). a) raw pseudo-forecast, b) post-processing with *DeFoReSt* and c) post-processing with *boosted recalibration*.





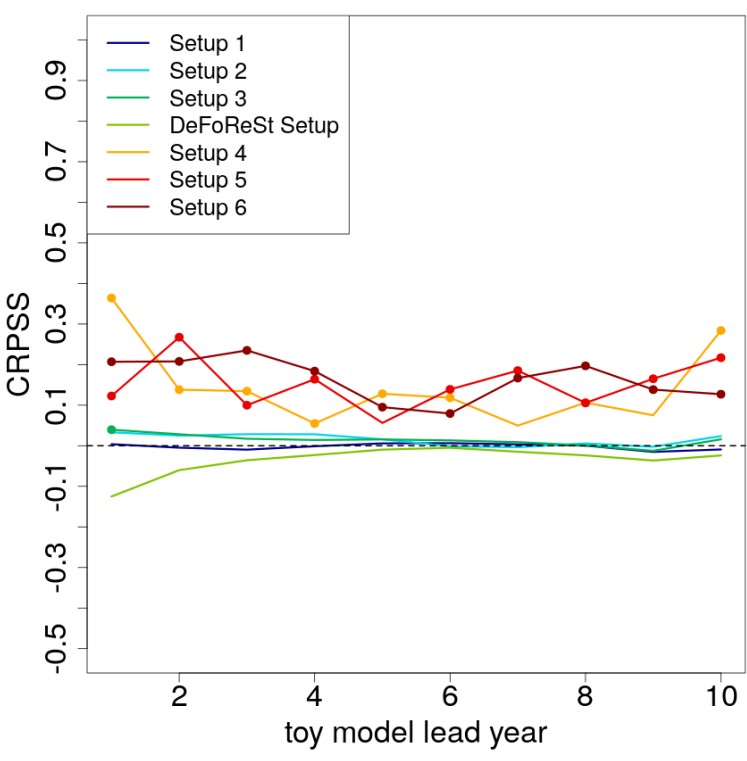

**Figure 6.** $CRPSS$ of different toy model setups with high potential predictability ($\eta = 0.8$, colored lines) post-processed with *boosted recalibration*. The associated toy model setups post-processed with *DeFoReSt* are used as reference for the skill-score. $CRPSS$ larger zero implies *boosted recalibration* performing better than *DeFoReSt*. Colored dots in Fig. 6 denote significance in the sense that the 0.025 and 0.975 quantiles from the 100 experiments do not include 0.





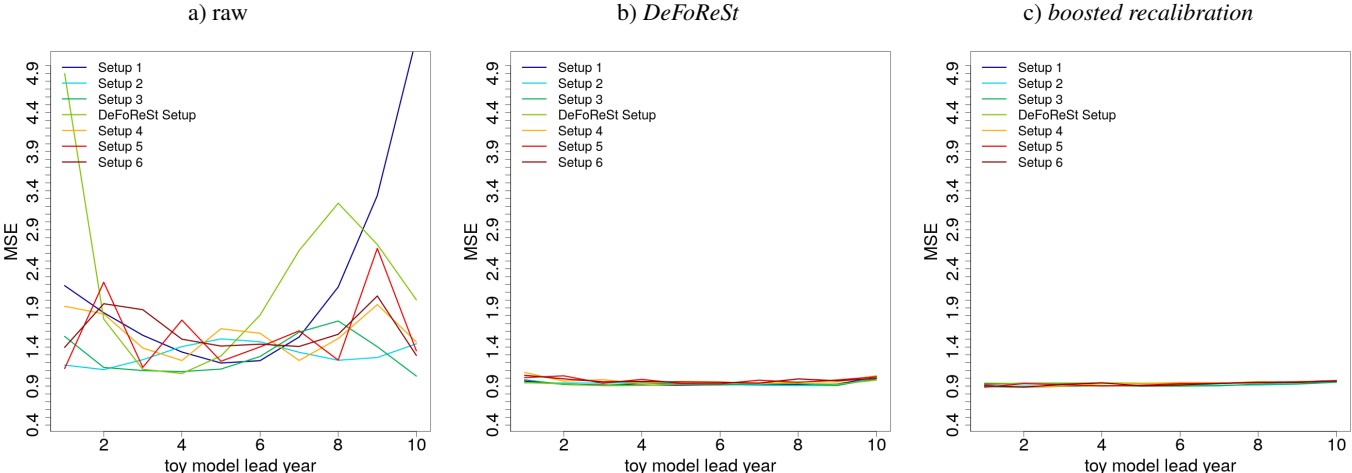

**Figure 7.** Mean squared error ($MSE$) of different toy model setups with low potential predictability ($\eta = 0.2$, colored lines). a) raw pseudo-forecast, b) post-processing with *DeFoReSt* and c) post-processing with *boosted recalibration*.





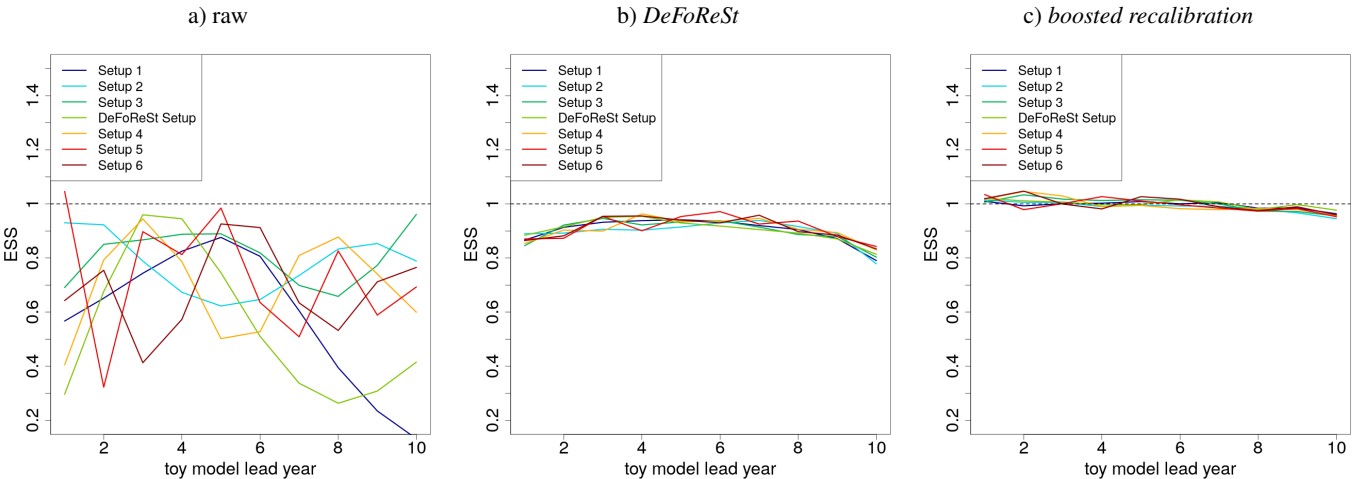

**Figure 8.** Ensemble spread score ($ESS$) of different toy model setups with low potential predictability ($\eta = 0.2$, colored lines). a) raw pseudo-forecast, b) post-processing with *DeFoReSt* and c) post-processing with *boosted recalibration*.





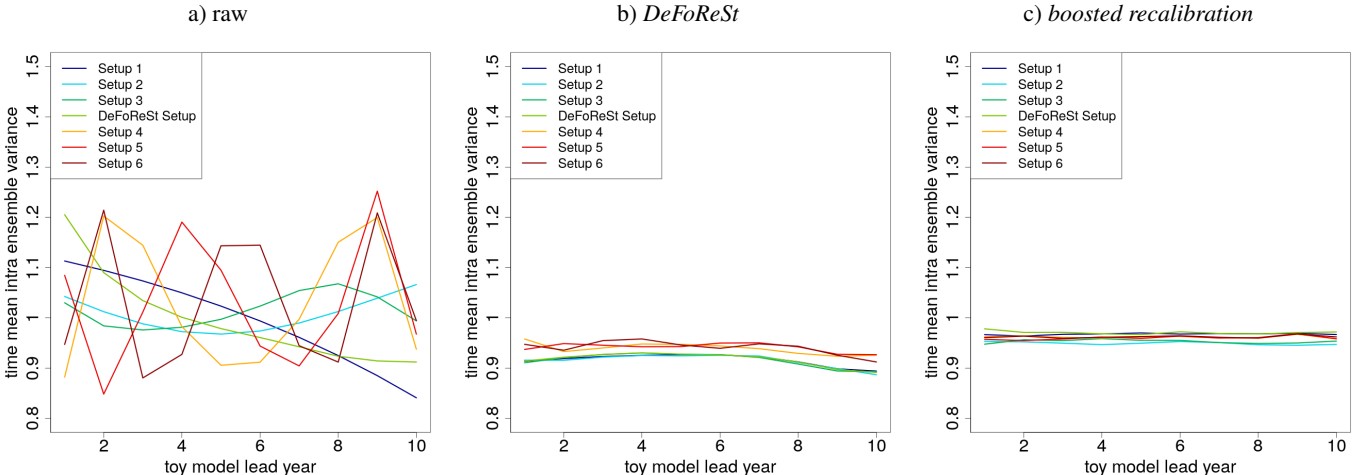

**Figure 9.** Intra ensemble variance (temporal average) of different toy model setups with low potential predictability ($\eta = 0.2$, colored lines).
a) raw pseudo-forecast, b) post-processing with *DeFoReSt* and c) post-processing with *boosted recalibration*.



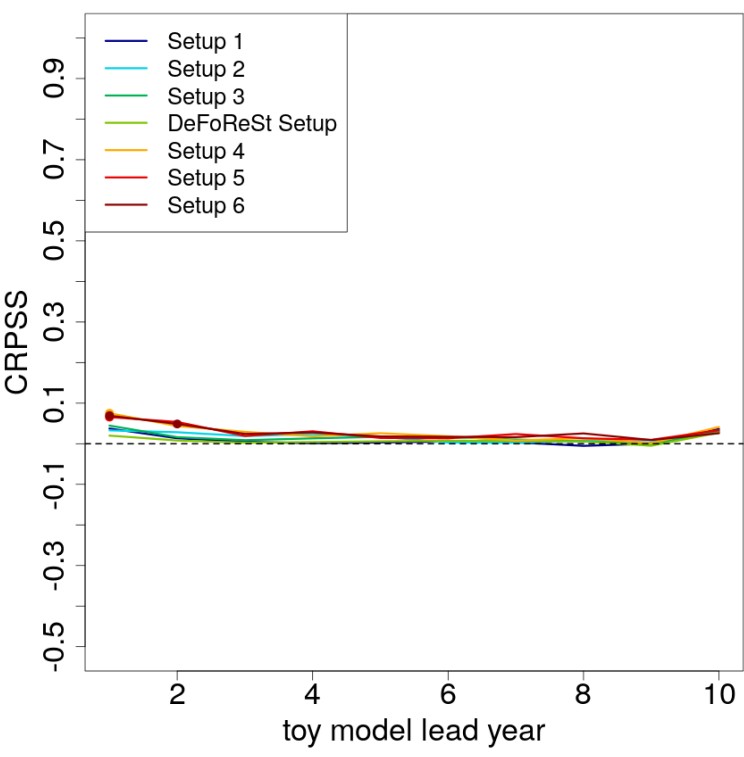

**Figure 10.** $CRPSS$ of different toy model setups with low potential predictability ($\eta = 0.2$, colored lines) post-processed with *boosted recalibration*. The associated toy model setups post-processed with *DeFoReSt* are used as reference for the skill-score. $CRPSS$ larger zero implies *boosted recalibration* performing better than *DeFoReSt*. Colored dots indicate lead years with either significant positive or negative values based on a 95% confidence interval from bootstrapping (100 repititions).





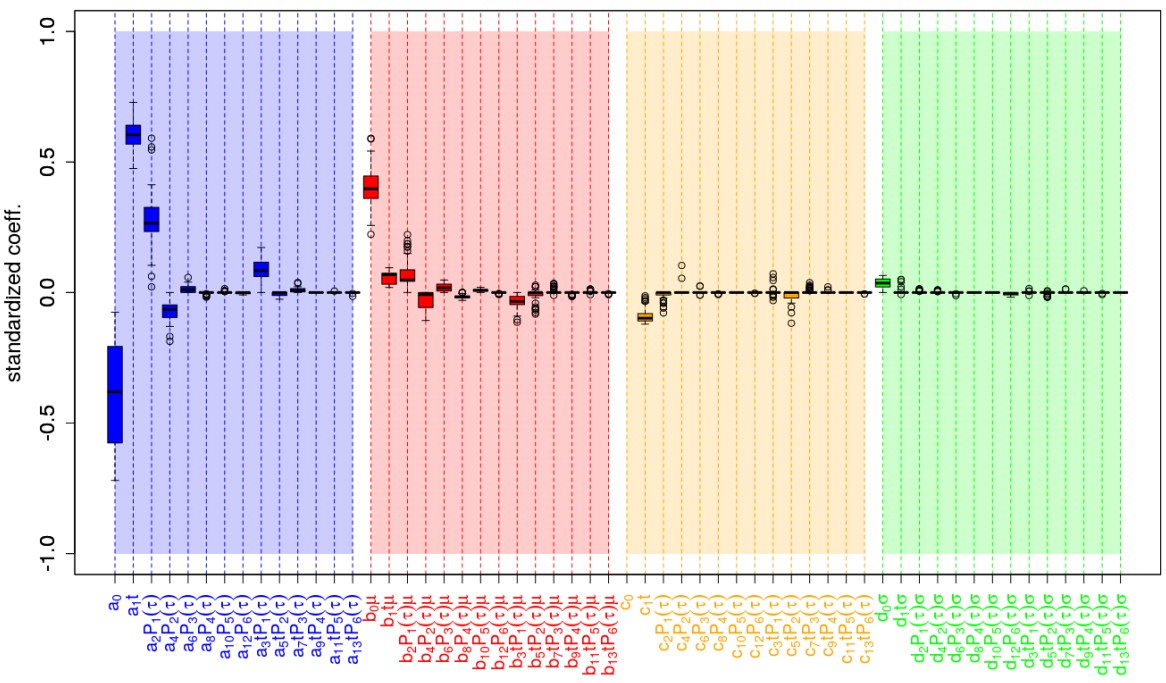

**Figure 11.** Coefficient estimates for recalibrating global mean 2m-Temperature of the *MiKlipl Prototype System*. Colored boxes represent the inter-quartile range ($IQR$) around the median (central, bold and black line) for coefficient estimates from the cross-validation setup; Whiskers denote maximum 1.5IQR. Coefficients are grouped accorting to correcting unconditional bias (blue), conditional bias (red), unconditional dispersion (orange) and conditional dispersion (green). Values refer to coefficients $a_0, b_0, c_0, d_0, ..., a_6, b_6, c_6, d_6$ and not to the product between these coefficients and the corresponding predictors (e.g. $a_2 P_1(\tau)$ refers to $a_2$). Please note, the value $c_0$ is around -2.5, but for a better overview the vertical axis is limited to the values range between -1 and 1. Vertical dashed bars highlight coefficients related to lead time dependent terms.





**Figure 12.** a) MSE, b) Reliability, c) Ensemble Variance and d) CRPSS of global mean surface temperature without any correction (black line), after recalibration with *DeFoReSt* (blue line) and *boosted recalibration* (red line). The CRPSS for the raw forecasts (black line) is for lead year 1 smaller than -1 and therefore not shown. The vertical bars show the 95% confidence interval due 1000-wise bootstrapping.



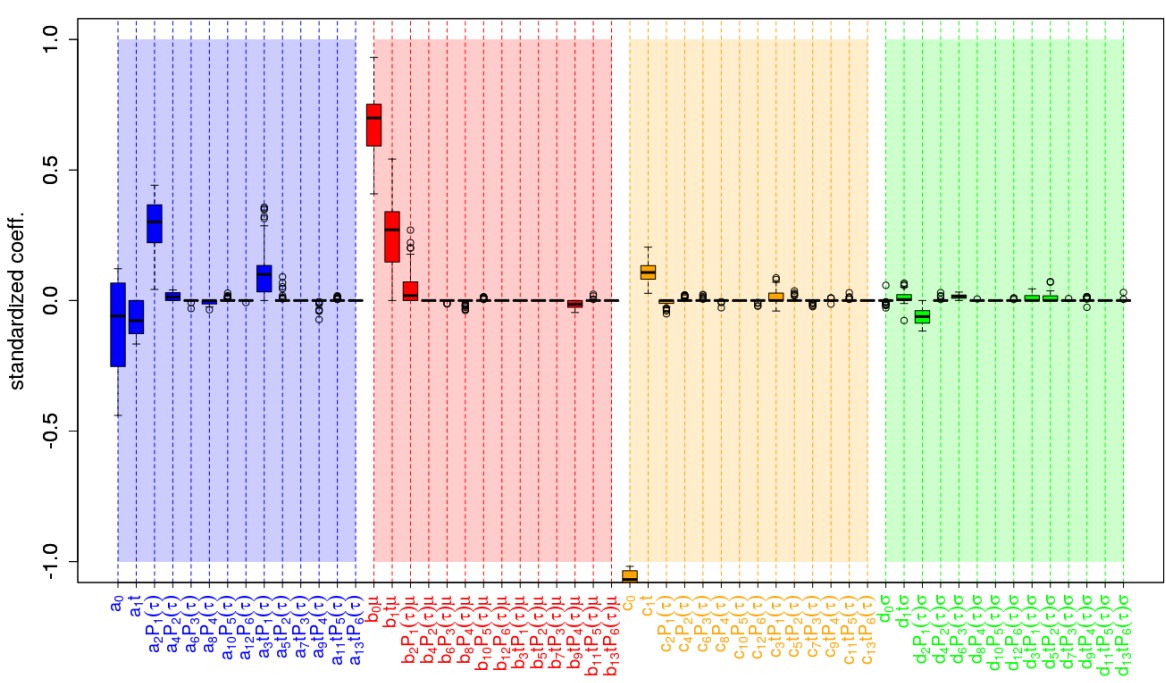

**Figure 13.** Identified coefficients for recalibrating the mean 2m-Temperature over the North Atlantic of prototype. Here, the coefficients are grouped by correcting uncond. bias (blue bars), cond. bias (red bars), uncond. dispersion (orange bars) and cond. dispersion (green bars). The coefficients are standardized, i.e. higher values implying a higher relevance. Values refer to coefficients $a_0, b_0, c_0, d_0, ..., a_6, b_6, c_6, d_6$ and not to the product between these coefficients and the corresponding predictors (e.g. $a_2 P_1(\tau)$ refers to $a_2$).







**Figure 14.** a) MSE, b) Reliability, c) Ensemble Variance and d) CRPSS of surface temperature over the North Atlantic without any correction (black line), after recalibration with *DeFoReSt* (blue line) and *boosted recalibration* (red line). The CRPSS for the raw forecasts (black line) is for lead year 1 smaller than -1 and therefore not shown. The vertical bars show the 95% confidence interval due 1000-wise bootstrapping.

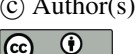



## Appendix A: Toy model construction

The toy model proposed by Pasternack et al. (2018) consists of pseudo-observations $x(t+\tau)$ and associated ensemble predictions, hereafter named pseudo-forecasts $f(t,\tau)$.

Both are based on an arbitrary but predictable signal $\mu_x$. Although almost identical to Pasternack et al. (2018), we quote the
construction of pseudo-observations in the following for purposes of overview.

The pseudo-observations $x$ is the sum of this predictable signal $\mu_x$ and an unpredictable noise term $\epsilon_x$,

$$x(t+\tau) = \mu_x(t+\tau) + \epsilon_x(t+\tau). \tag{A1}$$

Following Kharin et al. (2012) $\mu_x$ can be interpreted as the atmospheric response to slowly varying and predictable boundary conditions, while $\epsilon_x$ represents the unpredictable chaotic components of the observed dynamical system. $\mu_x$ and $\epsilon_x$ are assumed
to be stochastic Gaussian processes

$$\mu_x(t+\tau) \sim \mathcal{N}(0, \sigma_{\mu_x}^2) \qquad \text{with} \qquad \sigma_{\mu_x}^2 = \eta^2 \leq 1 \tag{A2}$$

and

$$\epsilon_x(t+\tau) \sim \mathcal{N}(0, \sigma_{\epsilon_x}^2) \qquad \text{with} \qquad \sigma_{\epsilon_x}^2 = 1 - \eta^2. \tag{A3}$$

The variation of $\mu_x$ around a slowly varying climate signal can be interpreted as the predictable part of decadal variability, its
amplitude is given by the variance $\mathrm{var}(\mu_x(t+\tau)) = \sigma_{\mu_x}^2$. The total variance of the pseudo-observations is thus $\mathrm{Var}(x) = \sigma_x^2 = \sigma_{\mu_x}^2 + \sigma_{\epsilon_x}^2$. Here, the relation of the latter two is uniquely controlled by the parameter $\eta \in [0,1]$, which can be interpreted as potential predictability ($\eta^2 = \sigma_{\mu_x}^2 / \sigma_x^2$).

In this toy model setup, the concrete form of this variability is not considered and thus taken as random. A potential climate trend could be superimposed as a time varying mean $\mu(t) = E[x(t)]$. As for the recalibration strategy only a difference in
trends is important, we use $\mu(t) = 0$ and $\alpha(t,\tau)$ addressing this difference in trends of forecast and observations.

The pseudo-forecast with ensemble members $f_i(t,\tau)$ for observations $x(t+\tau)$ is specified as:

$$f_i(t,\tau) = \mu_{ens}(t,\tau) + \epsilon_i(t,\tau), \tag{A4}$$

where $\mu_{ens}(t,\tau)$ is the ensemble mean and

$$\epsilon_i(t,\tau) \sim \mathcal{N}(0, \sigma_{ens}^2(t,\tau)) \tag{A5}$$

is the deviation of ensemble member $i$ from the ensemble mean; $\sigma_{ens}^2$ is the ensemble variance. In general, ensemble mean and ensemble variance both can be dependent on lead time $\tau$ and initialization time $t$. We relate the ensemble mean $\mu_{ens}(t,\tau)$ to the predictable signal in the observations $\mu_x(t,\tau)$ by assuming a) a systematic deviation characterized by an unconditional bias $\chi(t,\tau)$ (accounting also for a drift and difference in climate trends), a conditional bias $\psi(t,\tau)$ and b) a random deviation $\epsilon(t,\tau)$:

$$\mu_{ens}(t,\tau) = \chi(t,\tau) + \psi(t,\tau)\left(\mu_x(t,\tau) + \epsilon_f(t,\tau)\right), \tag{A6}$$





with $\epsilon_f(t,\tau) \sim \mathcal{N}(0, \sigma_{\epsilon_f}(t,\tau))$ being a random forecast error with variance $\sigma^2_{\epsilon_f}(t,\tau) < \sigma^2_{\epsilon_x}$. Although the variance of the random forecast error can in principle be dependent on lead time $\tau$ and initialization time $t$, we assume for simplicity a constant variance $\sigma^2_{\epsilon_f}(t,\tau) = \sigma^2_{\epsilon_f}$.

In contrast to the original toy model design, proposed by Pasternack et al. (2018), we assume an ensemble dispersion related to the variability of the unpredictable noise term $\epsilon_x$ with an unconditional and a conditional inflation factor ($\xi(t,\tau)$ and $\omega(t,\tau)$)

$$\sigma^2_{\mathrm{ens}}(t,\tau) = (\xi(t,\tau) + \omega(t,\tau)(\sigma_{\epsilon_x} - \sigma_{\epsilon_f}))^2. \tag{A7}$$

According to Eq. A6 the forecast ensemble mean $\mu_{\mathrm{ens}}$ is simply a function of the predictable signal $\mu_x$. In this toy model formulation, an explicit formulation of $\mu_x$ is not required, hence a random signal might be used for simplicity and it would

be legitimate to assume $E[\mu_x] = \mu(t+\tau) = 0$ without restricting generality. Here, we propose a linear trend in time $E[\mu_x] = \mu(t+\tau) = m_0 + m_1 t$ to emphasize a typical problem encountered in decadal climate prediction: different trends in observations and predictions (Kruschke et al., 2015).

   Given this setup, a choice of $\chi(t,\tau) \equiv 0$, $\psi(t,\tau) \equiv 1$, $\xi(t,\tau) \equiv 0$ and $\omega(t,\tau) \equiv 1$ would yield a perfectly calibrated ensemble forecast:

$$f^{\mathrm{perf}}(t,\tau) \sim \mathcal{N}(\mu_x(t,\tau), \sigma^2_{\epsilon_x}(t,\tau)). \tag{A8}$$

The ensemble mean $\mu_x(t,\tau)$ of $f^{\mathrm{perf}}(t,\tau)$ is equal to the predictable signal of the pseudo-observations. The ensemble variance $\sigma^2_{\epsilon_x}(t,\tau)$ is equal to the variance of the unpredictable noise term representing the error between the ensemble mean of $f^{\mathrm{perf}}(t,\tau)$ and the pseudo-observations. Hence, $f^{\mathrm{perf}}(t,\tau)$ is perfectly reliable.

   As mentioned in 4 this toy model setup is controlled on the one hand by $\eta$ characterizing the potential predictability and

on the other hand by $\chi(t,\tau)$, $\psi(t,\tau)$, $\xi(t,\tau)$ and $\omega(t,\tau)$, which control the unconditional and the conditional bias and the dispersion of the ensemble spread.

   Here, $\chi(t,\tau)$, $\psi(t,\tau)$, $\xi(t,\tau)$ and $\omega(t,\tau)$ are obtained from $\alpha(t,\tau)$, $\beta(t,\tau)$, $\gamma(t,\tau)$ and $\delta(t,\tau)$ as follows:

$$\chi(t,\tau) = -\frac{\alpha(t,\tau)}{\beta(t,\tau)} \tag{A9}$$

$$\psi(t,\tau) = \frac{1}{\beta(t,\tau)} \tag{A10}$$

$$\xi(t,\tau) = -\frac{\gamma(t,\tau)}{\delta(t,\tau)} \tag{A11}$$

$$\omega(t,\tau) = \frac{1}{\delta(t,\tau)}. \tag{A12}$$

The parameters $\chi(t,\tau)$, $\psi(t,\tau)$, $\xi(t,\tau)$ and $\omega(t,\tau)$ are defined such that a perfectly recalibrated toy model forecast $f^{\mathrm{Cal}}$ would have the following form:

$$f_i^{Cal}(t,\tau) \sim \mathcal{N}(\alpha(t,\tau) + \beta(t,\tau)\,\mu_{\mathrm{ens}}(t,\tau), (\exp(\gamma(t,\tau) + \delta(t,\tau)\,\sigma_{\mathrm{ens}}(t,\tau)))^2), \tag{A13}$$





Applying the definitions of $\mu_{\text{ens}}$ (Eq. A6) and $\sigma_{\text{ens}}$ (Eq. A7) leads to

$$f_i^{Cal}(t,\tau) \sim \mathcal{N}(\alpha(t,\tau) + \beta(t,\tau)(\chi(t,\tau) + \psi(t,\tau)\mu_x(t,\tau)), (\exp(\gamma(t,\tau) + \delta(t,\tau)(\xi(t,\tau) + \omega(t,\tau)\sigma_{\epsilon_x}(t,\tau))))^2), \tag{A14}$$

and applying the definitions of $\chi(t,\tau)$, $\psi(t,\tau)$ and $\omega(t,\tau)$ (Eqs. A9-A12) to (A14) would further lead to:

$$f_i^{Cal}(t,\tau) \sim \mathcal{N}(\alpha(t,\tau) - \beta(t,\tau)\frac{\alpha(t,\tau)}{\beta(t,\tau)} + \frac{\beta(t,\tau)}{\beta(t,\tau)}\mu_x(t,\tau), \frac{\gamma(t,\tau)}{\gamma(t,\tau)}\sigma_{\epsilon_x}^2(t,\tau)), \tag{A15}$$

This shows that $f^{\text{Cal}}$ is equal to the perfect toy model $f^{\text{Perf}}(t,\tau)$ (A8) :

$f^{\text{Cal}}(t,\tau) \sim \mathcal{N}(\mu_x(t,\tau), \sigma_{\epsilon_x}^2(t,\tau)). \tag{A16}$

This setting has the advantage that the perfect estimation of $\alpha(t,\tau)$, $\beta(t,\tau)$, $\gamma(t,\tau)$ and $\delta(t,\tau)$ is already known prior to calibration with minimization of the logarithmic likelihood.

As described in 3.2, a $6^{th}$ order polynomial approach was chosen for unconditional $\alpha(t,\tau)$, $\beta(t,\tau)$, $\gamma(t,\tau)$ and $\delta(t,\tau)$, yielding

$\alpha(t,\tau) = \sum_{l=0}^{6}(a_{2l} + a_{(2l+1)}t)P_l(\tau), \tag{A17}$

$\beta(t,\tau) = \sum_{l=0}^{6}(b_{2l} + b_{(2l+1)}t)P_l(\tau), \tag{A18}$

$\gamma(t,\tau) = \sum_{l=0}^{6}(c_{2l} + c_{(2l+1)}t)P_l(\tau), \tag{A19}$

$\delta(t,\tau) = \sum_{l=0}^{6}(d_{2l} + d_{(2l+1)}t)P_l(\tau). \tag{A20}$

For the current toy model experiment, we exemplarily specify values for $a_i$, $b_i$, $c_i$ and $d_i$ as obtained from calibrating the
MiKlip Prototype surface temperature over the North Atlantic against HadCRUT4 ($T_{obs}$):

$$E[T_{obs}] \sim \mathcal{N}(\alpha(t,\tau) + \beta(t,\tau)\bar{f}_{Prot}(t,\tau), (\exp(\gamma(t,\tau) + \delta(t,\tau)\sigma_{f_{\text{Prot}}}(t,\tau)))^2), \tag{A21}$$

where $\bar{f}_{Prot}$ and $\sigma_{f_{\text{Prot}}}$ specifying the corresponding ensemble mean and ensemble spread.

The values of the coefficients are given in Tab. A1.





|  | l=0 | l=1 | l=2 | l=3 | l=4 | l=5 | l=6 | l=7 | l=8 | l=9 | l=10 | l=11 | l=12 | l=13 |
|---|---|---|---|---|---|---|---|---|---|---|---|---|---|---|
| $a_l$ | -0.75 | 0.03 | 10.2 | 0.15 | -1.54 | -0.13 | 5.4 | -0.08 | -5 | 0.5 | -5 | 0.5 | -5 | 0.5 |
| $b_l$ | 0.67 | -0.0004 | 0.35 | -0.12 | 0.94 | 0.008 | 3.27 | -0.028 | 5 | -0.05 | 5 | -0.05 | 5 | -0.05 |
| $c_l$ | -0.79 | 0.03 | 9.62 | 0.18 | -0.93 | -0.16 | 5.74 | -0.08 | 5 | 0.5 | 5 | 0.5 | 5 | 0.5 |
| $d_l$ | 6.4 | 0.004 | -1.88 | -1.19 | 16.8 | 0.03 | 35.8 | -0.33 | 5 | 0.5 | 5 | 0.5 | 5 | 0.5 |

**Table A1.** Overview of the values coefficients $a_l$, $b_l$ and $w_l$.





*Author contributions.* Alexander Pasternack, Jens Grieger, Henning W. Rust and Uwe Ulbrich established the scientific scope of this study.

Alexander Pasternack, Jens Grieger and Henning W. Rust developed the algorithm of *boosted recalibration* and designed the toy model applied in this study. Alexander Pasternack carried out the statistical analysis and evaluated the results. Jens Grieger supported the analysis regarding post-processing of decadal climate predictions. Henning W. Rust supported the statistical analysis. Alexander Pasternack wrote the manuscript with contribution from all co-authors.

*Competing interests.* The authors declare that they have no conflict of interest.

*Acknowledgements.* This study was funded by the German Federal Ministry for Education and Research (BMBF) project MiKlip (sub-projects CALIBRATION Förderkennzeichen FKZ 01LP1520A.



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
