# Peer review of "Recalibrating Decadal Climate Predictions"

_Geoscientific Model Development, 2020_

## Referee Comment (RC1) · Anonymous Referee #1 · 19 Oct 2020

The manuscript builds on the post-processing procedure DeFoReSt proposed by Pasternack et al 2018 and presents a boosted re-calibration of decadal climate predictions. The manuscript describes a well thought approach on handling drift corrections and present it reasonable well for an statistical audience. The comparison between the boosted and the non-boosted calibration is excessive and described well, but lacks hypothesis testing to determine the actual differences between the two approaches outside of the argument that it is obvious.

While further work is required on the general presentation to make it more accessible to a wider audience, the authors might reconsider the choice of the journal, as the extreme focus on the statistical approach might be more appropriate for NPG. In its current shape the manuscript needs a much better illustration what has been done and

why it matters. Therefore, I recommend major revisions for the manuscript and would expect a rework of the figures and potentially the structure of the arguments.

Specific points:

18: "Significant advances could be achieved by recent progress in model development, data assimilation and climate observation." -> has been made

25: "unconditional, and conditional" -> unnecessary comma

37: "third/second" -> why third before second?

47: "objective function": objective has a specific meaning in statistics (see Jeffrey's prior) and would have to be individually proven. It is an unfortunate choice of word as it plays into the idea that statistics might be objective. As such, the word objective should be omitted in the manuscript completely.

87: "For the sake of completeness and readability these are presented in this section again." - Unnecessary sentence

124: By introducing the normal distribution with an calligraphic N and then use for the standard normal distribution greek letters, it gets quite confusing. As such this part needs to be rewritten. I would suggest to introduce N_S or similar for the standard normal distribution. As the authors work beforehand with large letters for CDFs, I would recommend to use a consistent approach for the nomenclature. I am aware that the equation for the CRPS is shown in this way often in statistical leaning literature, but as GMD is not such a journal I strongly recommend intuitive naming of variables.

138ff: I would strongly recommend a schematic on which basis the authors explain the mechanism of DeFoRFeSt. Equations are fine, but as they become extremely lengthy and hard to understand for the general reader (like eq 13), they need support and motivation.

185: Figure 1: name it consistent with Fig. 1 or rename all Figs to Figures.

202ff: The problem at this point is that the boosting algorithm forms an essential part for the understanding of the manuscript. I would strongly recommend the design of a schematic to make clear what exactly is done in the boosting process (apart from the equation, but the algorithmic strategy). This part of the manuscript needs effort to make it better understandable for the wider audience, especially as the authors do not publish here for a statistical, but a general model related audience.

202: "R-function poly" please make it a proper reference

205: "R-package crch" please make it a proper reference

206: "http://cran.r-project.org/" should go into the references

218: The way it is written the choice of nu requires a sensitivity test. So either it requires the motivation for choosing nu = 0.05 to be rewritten, or a demonstration and discussion of its effect.

226: The description of the cross-validation is not sufficient. A CV requires the statement on how the non-training data is afterwards evaluated (without taking into account the training data, otherwise it is not a CV but a Jackknife). The authors point to equation 21, but it is just the basis for the validation (which is described in line 216 with the Pearson correlation). So it would be required to state exactly what process is used for validation, which data is used for this step and which exact metric is applied to make the statement on a validated result.

238ff: Again the authors try in this section to explain everything by equations without explaining to the readers what consequences each of the decisions made have. The authors talk about extreme toy model experiments (l. 238), but do not state in what manner it is extreme. Then the authors introduce 5 parameters determining the experiments, but fail apart from short descriptions (like (un)conditional bias) to explain the reader what this actually means (and yes I am aware that most will know what it means in the direct community, but I think the authors should make the effort to explain it better

as it builds a foundation of their argument). So I would recommend here to create a figure explaining the consequences of each of the parameters to give the modelling community an entry point to follow the experiments to find analogues between the toy model and the usually used GCMs or similar (this has been done in Pasternack et al 2018, but perhaps a even more simplified/schematic version of Figure like Fig. 1 there will help). Giving the reader only an entry point by table 1 is not enough.

267ff: The authors show a very large figure with many elements in 4 main colours for the different parameters, but just spend three sentences without putting it in context and give the plot any meaning (e.g. comparison, interpretation apart from first three coefficients vs. last three). As such either the plot has not more information, then it is doubtful whether the plot has any use for the manuscript, or the many different whisker plots are important and it is not represented in the text. Just showing them is not enough, especially as later it is not referenced back to the figure when similar coefficient plots are made.

281: Estimating the 0.025 and 0.975 percentile from just 100 experiments is not a good way to demonstrate significances. The authors should either choose more experiments or go to alpha = 10. Or the description is so misunderstand-able that in fact more than 100 values to estimate the percentiles are used. In that case the section has to be rewritten.

283: (see 4) : What is referenced here?

285ff: Is there a reason, why in the DeFoReSt mode close to all metrics from Fig 3-10 show a U-shape over the lead years?

288: It is not explained why the uncertainties of the ESS are not visible (either small or not calculable).

330ff: Two consecutive sentences start with "Here,"

334 Why is there a bootstrapping in this section but not in the section above?

340ff: Why is there no comparison to the coefficients in Fig. 2?

348: "have also some impact." This should be analysed with a significance test and statements made accordingly

376: Are there significant differences between global and NA 2m-Temperature? Why is North Atlantic framed here as independent compared to the global and the comparison between those kept so short? It seems like it is written currently that one example would be sufficient. So why are the two not conclusively compared with each other in one section? So could there be a different story apart from just showing the statistical model applied to data?

Fig3-5 should be combined in one figure with 9 panels

Fig7-9 should be combined in one figure with 9 panels

Fig 6+10 potentially better to have them in one plot with 2 panels

Fig11: MiKlipl -> MiKlip

Fig12+14: Even when it is a stylistic choice: Why have the authors chosen a different colour-scheme compared to all the other figures in this manuscript?

---

## Referee Comment (RC2) · Anonymous Referee #2 · 26 Oct 2020

**1   General comments**

The authors present an extension to their previously introduced recalibration approach for decadal climate forecasts. The existing method is extended with a model selection approach using boosting to infer a parsimonious model from the data. Strengths and limitations of this approach are tested using synthetic data and an application to global mean and North Atlantic temperature forecasts is presented. While the boosting method presents a welcome addition to make the approach more generally useful across a diversity of applications (not limited to decadal forecasting) and therefore certainly merits publication, the article lacks in a few key aspects detailed below. Therefore, I suggest to accept the article subject to major revisions.

[Figure]

**1.1 Interpretation of the results**

The authors focus on descriptive verification measures to discuss the results from boosted recalibration. In addition, I suggest the authors expand the discussion of the inner workings of the method and the configuration that is identified as optimal with boosting. From a methods perspective, I wonder if the boosted recalibration models are of lower complexity compared with DeFoReSt (i.e. if boosting actually manages to efficiently constrain the number of parameters). Also, the selected models appear still quite complex given the limited data at hand to train these. Have you explored early stopping rules for the boosting approach (generally skill improves rapidly in the first iterations and levels out afterwards, potentially another criterion for stopping provides better generalization ability through reduced models)? From an application perspective, some more discussion on the identified nature of the error that is corrected with boosted recalibration would be useful, boosted recalibration is less effective if the systematic error has very simple structure as appears to be the case here.

**1.2 Link between the toy-model experiments and the application**

The authors quite clearly demonstrate the strengths and limitation of the boosted recalibration compared with the reference approach (DeFoReSt) using their toy model experiments. There is, however, no direct link drawn to the application of boosted recalibration with global mean and North Atlantic surface temperature forecasts. In particular, I would like to know if the lack of improvement from boosted recalibration compared with DeFoReSt is consistent with the adjustments that are applied (e.g. what errors are generally corrected).

**1.3 Significance assessment**

The significance assessment introduced on L280 does not reflect that the scores between DeFoReSt and boosted recalibration may be highly correlated due to the same forecast observation pairs being used. The 2.5-97.5% interval on the mean scores therefore likely underestimates the significance of the results. Instead, I propose to use a Diebold-Mariano test or a t-test on the score differences. I expect that using such a more powerful test would allow you to demonstrate e.g. that DeFoReSt significantly outperforms boosted recalibration when the error dependency matches the assumptions in DeFoReSt at least for short lead times.

**2  Minor comments**

L72: $1.5°$ and 40

L74: The full-field initialization

L151-2: the punctuation is somewhat weird, maybe this could be changed: "... drift adjusted ensemble mean forecast (i.e. a deterministic forecast without specific uncertainty quantification)."

L192-4: now is used three times

L209: Maybe mention that you chose maximum likelihood in the following for better readability.

L310: toy model setup with low potential predictability

L314: The ESS (see Fig. 8a-c) reveals that

L325: Typo? Shouldn't this read "the low predictability leads to a increased CRPS" (not reduced CRPSS)?

L332: Repetition, use "We discuss. . ." instead.

L337: Typo. 10-year validation period

L368: What fraction of the skill is due to the (linear) trend in global mean surface temperature?

L402: Pasternack et al. (2018) show that

L402: DeFoReSt leads to improved ensemble . . . or DeFoReSt leads to an improvement in ensemble . . .

L409-: Long sentence. Maybe start with "Common parameter estimation and model selection approaches such as stepwise regression and LASSO are designed for predictions of mean values. Non-homogeneous boosting jointly adjusts mean and variance and automatically. . . regression."

L423: this is not supported by your figure. Boosted recalibration is not (significantly) superior to DeFoReSt if errors are 'simple' according to Figure 6.

L438: equally

Figure 1: Why not show all the initialization times? The figure would be easily readable even with many more lines and the alignment of the differently colored blocks may become more apparent.

Fig. 3-5 and 7-9: Consider combining figures 3-5 and 7-9 each into one multi-panel plot to avoid splitting the figures across pages in the final publication. Also, the information shown is somewhat redundant and I encourage the authors to drop the sharpness plot for simplicity and for the following reasons: i) the sharpness of the raw model is of no use as it is not calibrated, ii) qualitative statements about the sharpness in DeFoReSt and boosted calibration can easily be derived from a visual comparison of the MSE and ESS plots. The legend should be shown only once for all 6 (9) panels of the multi-panel plot and axes should be labelled only once per row / column. Finally, consider

using a square-root (or log) transform on the y-axis to take away the focus from large differences with large scores.

Fig. 4: there is indication of extra over-confidence at the beginning and end of the forecast with DeFoReSt (with setups 1-3 and DeFoReSt). This appears to be an artefact of the method. Could you please discuss this?

Fig. 6, 10: Excessive whitespace. Please adjust the y-axis to better focus on the available data.
* * *

---

## Author Comment (AC1) · 15 Dec 2020

**Answer to referee 1**

Thank you very much for your informative and detailed comments.

**General comments**

"The manuscript builds on the post-processing procedure *DeFoReSt* proposed by Pasternack et al 2018 and presents a *boosted recalibration* of decadal climate predic-

tions. The manuscript describes a well thought approach on handling drift corrections and present it reasonable well for an statistical audience. The comparison between the boosted and the non-boosted calibration is excessive and described well, but lacks hypothesis testing to determine the actual differences between the two approaches outside of the argument that it is obvious.While further work is required on the general presentation to make it more accessible to a wider audience, the authors might reconsider the choice of the journal, as the extreme focus on the statistical approach might be more appropriate for NPG. In its current shape the manuscript needs a much better illustration what has been done and why it matters. Therefore, I recommend major revisions for the manuscript and would expect a rework of the figures and potentially the structure of the arguments."

**Specific points:**

1. 18: "Significant advances could be achieved by recent progress in model development,data assimilation and climate observation." → has been made

   **Answer:** Will be corrected

2. 25: "unconditional, and conditional" → unnecessary comma

   **Answer:** Will be corrected

3. 37: "third/second" → why third before second?

   **Answer:** third before second because the ensemble mean is corrected via a 3rd order polynomial and the ensemble spread via a second order polynomial, which is described a few sentences before (31ff).

4. 47: "objective function": objective has a specific meaning in statistics (see Jeffrey's prior) and would have to be individually proven. It is an unfortunate choice

of word as it plays into the idea that statistics might be objective. As such, the word objective should be omitted in the manuscript completely.

**Answer:** If the name "objective function" is misleading, we will change it to "cost function".

5. 87: "For the sake of completeness and readability these are presented in this section again." - Unnecessary sentence

   **Answer:** Will be deleted.

6. 124: By introducing the normal distribution with an calligraphic N and then use for the standard normal distribution greek letters, it gets quite confusing. As such this part needs to be rewritten. I would suggest to introduce $N_S$ or similar for the standard normal distribution. As the authors work beforehand with large letters for CDFs, I would recommend to use a consistent approach for the nomenclature. I am aware that the equation for the CRPS is shown in this way often in statistical leaning literature, but as GMD is not such a journal I strongly recommend intuitive naming of variables.

   **Answer:** We will replace the symbols $\Phi$ and $\varphi$ for the CDF and PDF of the standard normal distribution with $N_{S_C}$ and $N_{S_P}$.

7. 138ff: I would strongly recommend a schematic on which basis the authors explain the mechanism of DeFoRFeSt. Equations are fine, but as they become extremely lengthy and hard to understand for the general reader (like eq. 13), they need support and motivation.

   **Answer:** We will add such a schematic to the manuscript.

8. 185: Figure 1: name it consistent with Fig. 1 or rename all Figs to Figures.

   **Answer:** Will be corrected.

9. 202ff: The problem at this point is that the boosting algorithm forms an essential part for the understanding of the manuscript. I would strongly recommend the design of a schematic to make clear what exactly is done in the boosting process (apart from the equation, but the algorithmic strategy). This part of the manuscript needs effort to make it better understandable for the wider audience, especially as the authors do not publish here for a statistical, but a general model related audience.

   **Answer:** We will add a schematic flow chart describing the boosting algorithm analogously to Messner et al. 2017.

10. 202: "R-function poly" please make it a proper reference

    **Answer:** Will be corrected.

11. 205: "R-package crch" please make it a proper reference

    **Answer:** Will be corrected.

12. 206: "http://cran.r-project.org/" should go into the references

    **Answer:** Will be corrected.

13. 218: The way it is written the choice of nu requires a sensitivity test. So either it requires the motivation for choosing nu = 0.05 to be rewritten, or a demonstration and discussion of its effect.

    **Answer:** We will add a better motivation to the manuscript.

14. 226: The description of the cross-validation is not sufficient. A CV requires the statement on how the non-training data is afterwards evaluated (without taking into account the training data, otherwise it is not a CV but a Jackknife). The authors point to equation 21, but it is just the basis for the validation (which is described in line 216 with the Pearson correlation). So it would be required to

state exactly what process is used for validation, which data is used for this step and which exact metric is applied to make the statement on a validated result.

**Answer:** We will add a more detailed description.

15. 238ff: Again the authors try in this section to explain everything by equations without explaining to the readers what consequences each of the decisions made have. The authors talk about extreme toy model experiments (l. 238), but do not state in what manner it is extreme. Then the authors introduce 5 parameters determining the experiments, but fail apart from short descriptions (like (un)conditional bias) to explain the reader what this actually means (and yes I am aware that most will know what it means in the direct community, but I think the authors should make the effort to explain it better as it builds a foundation of their argument). So I would recommend here to create a figure explaining the consequences of each of the parameters to give the modelling community an entry point to follow the experiments to find analogues between the toy model and the usually used GCMs or similar (this has been done in Pasternack et al. 2018, but perhaps a even more simplified/schematic version of Figure like Fig. 1 there will help). Giving the reader only an entry point by table 1 is not enough

**Answer:** We will add a more detailed description and two figures showing the effect of the imposed systematic errors of the toy model. Moreover we will change the phrase '...we consider two extreme toy model experiments...' to '...we consider two toy model experiments with different potential predictabilities...'

16. 267ff: The authors show a very large figure with many elements in 4 main colours for the different parameters, but just spend three sentences without putting it in context and give the plot any meaning (e.g. comparison, interpretation apart from first three coefficients vs. last three). As such either the plot has not more information, then it is doubtful whether the plot has any use for the manuscript, or the many different whisker plots are important and it is not represented in the

text. Just showing them is not enough, especially as later it is not referenced back to the figure when similar coefficient plots are made.

**Answer:** Showing this Fig. 2 is relevant for the toy model construction since it supports the decision to use the same magnitude for the coefficients of the start and lead time dependent systematic errors. However, since it is not used for any further evaluations we will put it to the appendix related to the table A1 which shows the final coefficients for the toy model construction.

17. 281: Estimating the 0.025 and 0.975 percentile from just 100 experiments is not a good way to demonstrate significances. The authors should either choose more experiments or go to alpha = 10. Or the description is so misunderstandable that in fact more than 100 values to estimate the percentiles are used. In that case the section has to be rewritten.

**Answer:** Indeed, using 100 experiments is not enough for calculating the 0.025 and 0.975 percentile. We will repeat that with 1000 experiments and update the corresponding text passages and figures in the manuscript.

18. 283: (see 4) : What is referenced here?

**Answer:** Will be corrected

19. 285ff: Is there a reason, why in the *DeFoReSt* mode close to all metrics from Fig 3-10 show a U-shape over the lead years?

**Answer:** Regarding Figs. 3-10, particularly the ESS and the intra-ensemble variance omit a certain inverse U-shape. The reason might be, that *DeFoReSt* tends to be more underdispersive for the first and last lead year due to the missing additive correction term for the ensemble spread.

20. 288: It is not explained why the uncertainties of the ESS are not visible (either small or not calculable).

**Answer:** We have decided not to show any uncertainties for the ESS, since we just wanted to show the general effect of *boosted recalibration* and *DeFoReSt* and to ensure a better visibility.

21. 330ff: Two consecutive sentences start with "Here,".

    **Answer:** Will be corrected.

22. 334 Why is there a bootstrapping in this section but not in the section above?

    **Answer:** Unlike Sec. 4 we evaluate in Sec. 5 the CRPSS also w.r.t. a raw model. Thus, we decided to apply a bootstrapping approach to avoid any advantages of the post-processed models.

23. 334 Why is there a bootstrapping in this section but not in the section above?

    **Answer:** Unlike Sec. 4 we evaluate in Sec. 5 the CRPSS also w.r.t. a raw model. Thus, we decided to apply a bootstrapping approach to avoid any advantages of the post-processed models.

24. 340ff: Why is there no comparison to the coefficients in Fig. 2?

    **Answer:** The coefficients in Fig. 2 were used to derive the scale of the coefficients associated to 4th to 6th polynomials for the pseudo-forecasts. Here, unlike Fig. 11 and 13 no model selection was applied, i.e. a comparison is not very reasonable.

25. 348: "have also some impact." This should be analysed with a significance test and statements made accordingly

    **Answer:** We will change the statement "have also some impact" to "have also been identified by the boosting algorithm as relevant".

26. 376: Are there significant differences between global and NA 2m-Temperature? Why is North Atlantic framed here as independent compared to the global and the

comparison between those kept so short? It seems like it is written currently that one example would be sufficient. So why are the two not conclusively compared with each other in one section? So could there be a different story apart from just showing the statistical model applied to data?

**Answer:** *DeFoReSt* and *boosted recalibration* have been developed within MiKlip project. Here, the NA as well as the global 2m-temperature are the key variables within this project. Moreover these regions distinguish themselves by their potential predictability. Thus analog to the toy model experiments we show the mechanisms of theses recalibration approaches to MiKlip predictions with smaller and higher potential predictability. Furthermore, regarding the different identified predictor variables for the NA and global 2m-temperature (Figs. 11 and 13) one can see that other processes are relevant due to a different spatial scale of these examples.

27. Fig3-5 should be combined in one figure with 9 panels

    **Answer:** Will be corrected.

28. Fig7-9 should be combined in one figure with 9 panels

    **Answer:** Will be corrected.

29. Fig 6+10 potentially better to have them in one plot with 2 panels

    **Answer:** We would like to keep these plots separate, since they are discussed in different sections. Thus, to ensure a better readability it may be better to show these figures separately.

30. Fig11: MiKlipl → MiKlip

    **Answer:** Will be corrected

31. Fig12+14: Even when it is a stylistic choice: Why have the authors chosen a different colour-scheme compared to all the other figures in this manuscript?

**Answer:** We wanted to distinguish the toy model results optically from the results based on MiKlip data.

---

## Author Comment (AC2) · 15 Dec 2020

**Answer to referee 2**

Thank you very much for your informative and detailed comments.

**1. General comments**

"The authors present an extension to their previously introduced recalibration approach for decadal climate forecasts. The existing method is extended with a model selection approach using boosting to infer a parsimonious model from the data. Strengths and limitations of this approach are tested using synthetic data and an application to global mean and North Atlantic temperature forecasts is presented. While the boosting method presents a welcome addition to make the approach more generally useful across a diversity of applications (not limited to decadal forecasting) and therefore certainly merits publication, the article lacks in a few key aspects detailed below. Therefore, I suggest to accept the article subject to major revisions."

1.1 Interpretation of the results

"The authors focus on descriptive verification measures to discuss the results from *boosted recalibration*. In addition, I suggest the authors expand the discussion of the inner workings of the method and the configuration that is identified as optimal with boosting. From a methods perspective, I wonder if the *boosted recalibration* models are of lower complexity compared with *DeFoReSt* (i.e. if boosting actually manages to efficiently constrain the number of parameters). Also, the selected models appear still quite complex given the limited data at hand to train these. Have you explored early stopping rules for the boosting approach (generally skill improves rapidly in the first iterations and levels out afterwards, potentially another criterion for stopping provides better generalization ability through reduced models)? From an application perspective, some more discussion on the identified nature of the error that is corrected with *boosted recalibration* would be useful, *boosted recalibration* is less effective if the systematic error has very simple structure as appears to be the case here."

**Answer:** The basic feature of the boosting algorithm is to allow a priori for a complex structure of the model used for recalibration but use the complexity only as needed. Thus our procedure is able to adjust complexity according to the problem at hand based on out-of-sample prediction error. This is realized by the automatic selection of the most relevant predictor variables by iteratively updating the log-likelihood. For

each iteration step only one coefficient (the one that improves the fit most) is updated and thus complexity is successively increased. Here, the maximum number of iteration steps must be specified beforehand. However, if the chosen iteration step is small enough certain model coefficients are remaining zero. In order to find the best performing model an adequate iteration step has to be identified (model selection step) using a cross-validation setup. For this purpose we split the data into 5 parts and for each part, recalibrated predictions are computed from boosting model at the corresponding iteration step that were fitted on the remaining 4 parts. Afterwards the log-likelihood over all 5 recalibrated parts were summed up. This procedure is repeated for every iteration step. The iteration step with the lowest log-likelihood is considered as the one which provides the statistical model with the best predictive performance. Due to this procedure predictor variables of the statistical model that are not relevant are remaining zero. This can be seen in Figs. 11 and 13 which demonstrate which predictor variables are identified as relevant. Here, one can see that both for the North Atlantic as well as for the global 2m-temperature the complexity of *boosted recalibration* is around 15 identified predictor variables whereas *DeFoReSt* uses 22 predictor variables. We will add a schematic overview of the boosting algorithm and further explanation of the cross-validation approach to the manuscript.

1.2 Link between the toy-model experiments and the application

"The authors quite clearly demonstrate the strengths and limitation of the *boosted recalibration* compared with the reference approach (*DeFoReSt*) using their toy model experiments. There is, however, no direct link drawn to the application of *boosted recalibration* with global mean and North Atlantic surface temperature forecasts. In particular, I would like to know if the lack of improvement from *boosted recalibration* compared with *DeFoReSt* is consistent with the adjustments that are applied (e.g. what errors are generally corrected)."

**Answer:** With the toy model experiments we show that *boosted recalibration* outper-forms *DeFoReSt*, if the polynomial order of the systematic errors goes beyond the restrictions of the *DeFoReSt* design. If that is not the case, both recalibration methods perform equally. Regarding the global mean and North Atlantic surface temperature forecasts one can see in Figs. 11 and 13 that *boosted recalibration* mostly identified predictor variables with a polynomial order smaller than 3. Thus, the fact that *De-FoReSt* and *boosted recalibration* perform equally for recalibrating MiKlip temperature forecasts is in accordance to the toy model results. We will emphasize the connection between toy model and temperature results more in the manuscript.

1.3 Significance assessment

"The significance assessment introduced on L280 does not reflect that the scores be-tween *DeFoReSt* and *boosted recalibration* may be highly correlated due to the same forecast observation pairs being used. The 2.5-97.5% interval on the mean scores therefore likely underestimates the significance of the results. Instead, I propose to use a Diebold-Mariano test or a t-test on the score differences. I expect that using such a more powerful test would allow you to demonstrate e.g. that *DeFoReSt* sig-nificantly outperforms *boosted recalibration* when the error dependency matches the assumptions in *DeFoReSt* at least for short lead times."

**Answer:** Actually, we do not expected that *DeFoReSt* outperforms *boosted recalibra-tion*, because the systematic error in the Miklip data is unknown and therefore does not have to be equal to the *DeFoReSt*-scenario. *Boosted recalibration* is able to cover systematic errors up to the 6th polynomial order, which also includes the the *DeFoR-eSt*-scenario, but is more flexible due to boosting. One can see in Fig. 11 and 13 that the identified polynomials do not go beyond the 3rd order, which is caught by *DeFoReSt* just as well. To compare these two post-processing methods we applied a bootstrapping approach. Within the applied bootstrapping approach, we calculate

the score 1000 times, each with a different sample (replacements are allowed) from the original time series. The corresponding samples for the scores of *DeFoReSt* and *boosted recalibration* are not the same, i.e. a correlation between these scores is avoided. However, if these scores would base each in the same sample a high correlation between those is possible and a Diebold-Mariano test or a t-test would be meaningful, indeed. We will point this out more clearly in the manuscript.

**2. Minor comments**

1. L72: 1.5°and 40

   **Answer:** Will be corrected.

2. L74: The full-field initialization

   **Answer:** Will be corrected.

3. L151-2: the punctuation is somewhat weird, maybe this could be changed: "...drift adjusted ensemble mean forecast (i.e. a deterministic forecast without specific uncertainty quantification)."

   **Answer:** Will be corrected.

4. L192-4: now is used three times

   **Answer:** Will be corrected.

5. L209: Maybe mention that you chose maximum likelihood in the following for better readability.

   **Answer:** Will be corrected.

6. L310: toy model setup with low potential predictability

   **Answer:** Will be corrected.

7. L314: The ESS (see Fig. 8a-c) reveals that

    **Answer:** Will be corrected.

8. L325: Typo? Shouldn't this read "the low predictability leads to a increased CRPS" (not reduced CRPSS)?

    **Answer:** Actually not. In a setup with low potential predictability the benefit of *boosted recalibration* over *DeFoReSt* is smaller compared to a setup with high potential predictability. Thus the CRPSS is reduced.

9. L332: Repetition, use "We discuss..." instead

    **Answer:** Will be corrected.

10. L337: Typo. 10-year validation period

    **Answer:** Will be corrected.

11. L368: What fraction of the skill is due to the (linear) trend in global mean surface temperature?

    **Answer:** This is a very interesting question, indeed. It not possible to answer this briefly. We are currently working on a study where we use a recalibrated climatology as reference for the skill evaluation. The purpose is to analyze to what extent the predictive skill of recalibrated decadal predictions is superior to a statistical model with the same statistical properties as the applied recalibration strategy.

12. L402: Pasternack et al. (2018) show that

    **Answer:** Will be corrected.

13. L402: *DeFoReSt* leads to improved ensemble...or *DeFoReSt* leads to an improvement in ensemble...

    **Answer:** Will be corrected.

14. L409-: Long sentence. Maybe start with "Common parameter estimation and model selection approaches such as stepwise regression and LASSO are designed for predictions of mean values. Non-homogeneous boosting jointly adjusts mean and variance and automatically...regression."

    **Answer:** Will be corrected.

15. L423: this is not supported by your figure. *Boosted recalibration* is not (significantly)superior to *DeFoReSt* if errors are 'simple' according to Figure 6.

    **Answer:** Will be corrected.

16. L438: equally

    **Answer:** Will be corrected.

17. Figure 1: Why not show all the initialization times? The figure would be easily readable even with many more lines and the alignment of the differently colored blocks may become more apparent.

    **Answer:** We will replace that figure with an new one showing all initialization times.

18. Fig. 3-5 and 7-9: Consider combining figures 3-5 and 7-9 each into one multi-panel plot to avoid splitting the figures across pages in the final publication. Also, the information shown is somewhat redundant and I encourage the authors to drop the sharpness plot for simplicity and for the following reasons: i) the sharpness of the raw model is of no use as it is not calibrated, ii) qualitative statements about the sharpness in *DeFoReSt* and *boosted calibration* can easily be derived from a visual comparison of the MSE and ESS plots. The legend should be shown only once for all 6 (9) panels of the multi-panel plot and axes should be labelled only once per row / column. Finally, consider using a square-root (or log) transform on the y-axis to take away the focus from large differences with large scores.

**Answer:** Will be corrected. However, we still would like to keep the sharpness figures. Indeed one could derive the sharpness from the ESS and the MSE but we think that is may be more convenient to have a visual impression of the sharpness.

19. Fig. 4: there is indication of extra overconfidence at the beginning and end of the forecast with DeFoReSt (with setups 1-3 and *DeFoReSt*). This appears to be an artefact of the method. Could you please discuss this?

    **Answer:** Regarding the ESS of the raw model, one can see that for lead year 1 and 10 particularly the setups 1-3 are strongly over- or underconfident. Thus we would explain the inverse U-shape of the pseudo-forecasts after recalibration with *DeFoReSt* with the fact that *DeFoReSt* tends to be more underdispersive for the first and last lead year due to the missing additive correction term for the ensemble spread. This example shows that *boosted recalibration* can account better for forecasts which are either strongly overdispersive or strongly underdispersive.

20. Fig. 6, 10: Excessive white space. Please adjust the y-axis to better focus on the available data.

    **Answer:** Will be corrected.

---

## Editor Decision (ED1)

This manuscript, "Recalibrating Decadal Climate Predictions – What is an adequate model for the drift?" by Pasternack et al. (gmd-2020-191), makes important contribution to the field of bias correction (or adjustment) of near-term climate predictions. Although proposed novel methodology did not lead to substantial improvements in presently analysed fields over the pervious methods, clear exposition of the material put forward holds a potential that further more diverse application could reveal more benefit. I would like to see this paper published in GMD after the authors address few minor points listed below.

Lines 1-2: I would suggest: ".. such as multi-year to decadal forecasts are increasingly being used to guide adaptation measures and building of resilience."

Line 2: I would suggest: "To ensure the utility of multi-member probabilistic predictions, …"

Line 3: I would suggest: "be corrected or at least reduced."

Line 3: I would suggest: ".., such as the long-term forecast horizon, the …"

Line 6: I would suggest: "typical pairs of hindcasts and observations are available .."

Line 17: I would suggest: ".. of initialized forecasts of the climate for the coming years."

Line 18: I would suggest: ".. in model development, data assimilation for initialization and climate observations."

Line 19: I would suggest: ".. near-term climate information and services for adaptation and .."

Line 20. I would suggest: ".. DCPP and WCRP Grand Challenge on Near-Term Climate Prediction) and national .."

Line 22. I would suggest: "Typically, ensemble climate predictions are framed …"

Line 26: Add some relevant references here: ".. state towards its own climatology (Maraun, 2016, Fuckar et al., 2014)."

Maraun, D. (2016), Bias Correcting Climate Change Simulations - a Critical Review, *Curr Clim Change Rep* **2,** 211–220 (2016). https://doi.org/10.1007/s40641-016-0050-x

Fuckar, N. S., D. Volpi, V. Guemas, and F. J. Doblas-Reyes (2014), A posteriori adjustment of near-term climate predictions: Accounting for the drift dependence on the initial conditions, Geophys. Res. Lett., 41, 5200–5207, https://doi.org/10.1002/2014GL060815.

Line 88: For the benefit of a wider audience add here a general forecast verification reference: ".. and the verifying observations (Jolliffe and Stephenson, 2012)."

Ian T. Jolliffe, and David B. Stephenson (2012), Forecast Verification – A Practitioner's Guide in Atmospheric Science, 2nd ed., Wiley-Blackwell, 274 pp.

---

## Author Response (AR2)

**Answer to referee 1**

Thank you very much for your comments.

**General comments**

"The corrections proposed are sufficient to overcome the general reservations put forward in the first review. The manuscript has greatly improved and the three schematics included will hopefully help a wider audience to better understand the manuscript. Fig. 1 is certainly the hardest to understand and requires some reading of the caption by the reader, but as I do not have a better suggestion how to illustrates what happens in this section, it should be sufficient. Apart from some minor corrections I recommend the manuscript for publication"

**Specific points:**

1. 117: "Inconsistency $F_0$ *in eq. 4 and* $F_o$ *in text*"

   **Answer:** corrected

2. 205: "available from http://cran.r-project.org/ (CRAN)" not necessary"

   **Answer:** corrected

**Answer to referee 2**

Thank you very much for your comments.

**1. General comments**

"TThis manuscript, "Recalibrating Decadal Climate Predictions – What is an adequate model for the drift?" by Pasternack et al. (gmd-2020-191), makes important contribution to the field of bias correction (or adjustment) of near-term climate predictions. Although proposed novel methodology did not lead to substantial improvements in presently analysed fields over the pervious methods, clear exposition of the material put forward holds a potential that further more diverse application could reveal more benefit. I would like to see this paper published in GMD after the authors address few minor points listed below:"

**2. Minor comments**

1. Lines 1-2: I would suggest: ".. such as multi-year to decadal forecasts are increasingly being used to guide adaptation measures and building of resilience."

   **Answer:** corrected.

2. Line 2: I would suggest: "To ensure the utility of multi-member probabilistic predictions, . . ."

   **Answer:** corrected.

3. Line 3: I would suggest: "be corrected or at least reduced."

   **Answer:** corrected.

4. Line 3: I would suggest: ".., such as the long-term forecast horizon, the . . ."

   **Answer:** corrected.

5. Line 6: I would suggest: "typical pairs of hindcasts and observations are available .."

   **Answer:** corrected.

6. Line 17: I would suggest: ".. of initialized forecasts of the climate for the coming years."

   **Answer:** corrected .

7. Line 18: I would suggest: ".. in model development, data assimilation for initialization and climate observations."

   **Answer:** corrected.

8. Line 19: I would suggest: ".. near-term climate information and services for adaptation and .."

   **Answer:** corrected.

9. Line 20. I would suggest: ".. DCPP and WCRP Grand Challenge on Near-Term Climate Prediction) and national .."

   **Answer:** Will be corrected.

10. Line 22. I would suggest: "Typically, ensemble climate predictions are framed . . . "

    **Answer:** corrected.

11. Line 26: Add some relevant references here: ".. state towards its own climatology (Maraun, 2016, Fuckar et al., 2014)."

    **Answer:** corrected.

12. Line 88: For the benefit of a wider audience add here a general forecast verification reference: ".. and the verifying observations (Jolliffe and Stephenson, 2012)."

    **Answer:** corrected.

[revised manuscript text omitted]